# Progressively shifting patterns of co-modulation among premotor cortex neurons carry dynamically similar signals during action execution and observation

**Zhonghao Zhao[1], Marc H Schieber[1,2,3]\***

[1]Department of Biomedical Engineering, University of Rochester, Rochester, United States; [2]Department of Neurology, University of Rochester, Rochester, United States; [3]Department of Neuroscience, University of Rochester, Rochester, United States

## eLife Assessment

This **valuable** study reports on the characteristics of premotor cortical population activity during the execution and observation of a moderately complex reaching and grasping task. By using new variants of well-established techniques to analyse neural population activity, the authors provide **solid** evidence that while the geometry of neural population activity changes between execution and observation, their dynamics are largely preserved. Although these findings are novel and robust, pending additional controls and analyses, the authors should further clarify the functional implications of their findings.

***\*For correspondence:**
mschiebe@ur.rochester.edu

**Abstract** Neurons in macaque premotor cortex show firing rate modulation whether the subject performs an action or observes another individual performing a similar action. Although such mirror neurons have been thought to have highly congruent discharge during execution and observation, many, if not most, show noncongruent activity. Studies of reaching movements, for which low-dimensional neural trajectories exhibit comparatively simple dynamical motifs, have shown that these prevalent patterns of co-modulation pass through subspaces which are shared in part, but in part are visited exclusively during either execution or observation. The neural dynamics of hand movements are more complex, however. We developed a novel approach to examine prevalent patterns of co-modulation during execution and observation of a task that involved reaching, grasping, and manipulation. Rather than following neural trajectories in subspaces that contain their entire time course, we identified time series of instantaneous subspaces, calculated principal angles among them, sampled trajectory segments at the times of selected behavioral events, and projected those segments into the time series of instantaneous subspaces. These instantaneous neural subspaces most often remained distinct during execution versus observation. Nevertheless, latent dynamics during execution and observation could be partially aligned with canonical correlation, indicating some similarity of the relationships among neural representations of different movements relative to one another during execution and observation. We also found that during action execution, mirror neurons showed consistent patterns of co-modulation both within and between sessions, but other non-mirror neurons that were modulated only during action execution and not during observation showed considerable variability of co-modulation.

## Introduction

Although the premotor (PM) and primary motor cortex (M1) are generally thought to be involved in the planning and execution of movement, many neurons in these areas have been found to discharge not only when the subject executes a movement, but also when the subject observes a similar movement being performed by another individual. Such neurons have been found in the ventral premotor cortex (PMv) (*Maranesi et al., 2014*; *Gallese et al., 1996*), dorsal premotor cortex (PMd) (*Cisek and Kalaska, 2004*; *Papadourakis and Raos, 2019*; *Albertini et al., 2021*; *Pezzulo et al., 2022*), and M1 (*Dushanova and Donoghue, 2010*; *Kraskov et al., 2014*; *Vigneswaran et al., 2013*). The prevalence of such execution/observation neurons in cortical motor areas argues against their activity during observation being merely an epiphenomenon unrelated to their activity during execution but also poses a larger question: What is the nature of the relationship between their activity during execution versus observation?

Early studies of these neurons emphasized those with congruent discharge during execution and observation contexts. Congruent neurons discharged during the same type of grasp (*Gallese et al., 1996*; *Rizzolatti et al., 1996*) or retained the same preferred direction (*Dushanova and Donoghue, 2010*; *Kilner and Lemon, 2013*) during both execution and observation. Emphasis on such congruent neurons led to the notion that they mediate understanding of observed actions as they mirror their own activity during execution (*di Pellegrino et al., 1992*; *Rizzolatti and Craighero, 2004*).

In addition to congruent neurons, however, even early studies also reported many other noncongruent neurons that also discharged during execution and during observation but discharged differently in the two contexts (*Gallese et al., 1996*). In many studies, roughly half or more of the neurons modulated during both execution and observation were noncongruent (*Dushanova and Donoghue, 2010*; *Kraskov et al., 2014*; *Mazurek et al., 2018*; *Jiang et al., 2020*). Of PMv neurons modulated during both execution and observation, over the time course of behavioral trials, only ~20% showed brief periods with strictly congruent firing rates (*Pomper et al., 2023*). And in both PMv and PMd, the proportion of congruent neurons may not be different from that expected by chance alone (*Papadourakis and Raos, 2019*). Though many authors apply the term mirror neurons (MNs) strictly to highly congruent neurons, here, we will refer to all neurons modulated during both contexts—execution and observation—as MNs.

That so many MNs are active differently during action execution versus observation calls into question not only the extent to which the representation of movements by these neuron populations actually matches in the two contexts, but also the extent to which MN activity during observation has any meaningful function for the organism (*Hickok, 2009*; *Krakauer et al., 2017*). Nevertheless, multiple studies have found that of the neurons in cortical motor areas that are modulated during execution, a large fraction are also modulated during observation. For example, 31 of 64 (49%) pyramidal tract neurons in PMv and 65 of 132 (49%) in M1 showed modulation during both execution and observation (*Kraskov et al., 2009*; *Vigneswaran et al., 2013*; *Kraskov et al., 2014*). Such findings suggest that the observation-related activity of execution-related neurons in PMv, PMd, and M1, some of which project to the spinal cord, is somehow related to the motoric functions of these cortical areas.

The widely varying degrees of congruence versus noncongruence among individual MNs may obscure population-level relationships between their patterns of co-modulation during execution and observation. Behavior evolving in time may be represented more accurately by the temporal progression of co-modulation in populations of neurons than by the temporal pattern of firing rate in single neurons (*Shenoy et al., 2013*; *Cunningham and Yu, 2014*; *Vyas et al., 2020*). Patterns of co-modulation can be considered in a high-dimensional neural-state space where the firing rate of each neuron is a separate, orthogonal dimension. The instantaneous, simultaneous firing rates of all $N$ neurons then form a point in this space, and the time series of these instantaneous points traces out a neural trajectory over time. Neural population trajectories do not visit all regions of the $N$-dimensional state space equivalently, however. Dimensionality reduction techniques can be used to identify a small set of latent dimensions—a subspace—that captures the most prevalent patterns of co-modulation among the population of $N$ neurons.

Studies of neural trajectories underlying action execution that focused on reaching movements made with the arm have revealed that rotational motifs in a low-dimensional subspace capture much of the neural population's firing rate variance (*Churchland et al., 2012*; *Russo et al., 2020*). But the M1 neural trajectories underlying grasping movements (*Suresh et al., 2020*) or force production at

the wrist (*Dekleva et al., 2024*) are more complex. The latent subspaces that capture the predominant patterns of co-modulation among M1 neurons, for example, shift progressively over the time course of behavioral trials involving reaching for, grasping, and manipulating various objects at various locations (*Rouse and Schieber, 2018*).

A relevant but often overlooked aspect of such dynamics in neuron populations active during both execution and observation has to do with the distinction between condition-independent and condition-dependent variation in neuronal activity (*Kaufman et al., 2016*; *Rouse and Schieber, 2018*). The variance in neural activity averaged across all the conditions in a given task context is condition-independent. For example, in an eight-direction center-out reaching task, averaging a unit's firing rate as a function of time across all eight directions may show an initially low firing rate that increases prior to movement onset, peaks during the movement, and then declines during the final hold, irrespective of the movement direction. Subtracting this condition-independent activity from the unit's firing rate during each trial gives the remaining variance, and averaging separately across trials in each of the eight directions then averages out noise variance, leaving the condition-dependent variance that represents the unit's modulation among the eight directions (conditions). Alternatively, condition-independent, condition-dependent, and noise variance can be partitioned through demixed principal component analysis (PCA) (*Kobak et al., 2016*; *Gallego et al., 2018*). The extent to which neural dynamics occur in a subspace shared by execution and observation versus subspaces unique to execution or observation may differ for the condition-independent versus condition-dependent partitions of neural activity. Here, we tested the hypothesis that the condition-dependent activity of PM MN populations progresses through distinct subspaces during execution versus observation, which would indicate distinct patterns of co-modulation among MNs during execution versus observation.

Because of the complexity of condition-dependent neural trajectories for movements involving the hand, we developed a novel approach. Rather than examining trajectories over the entire time course of behavioral trials, we identified time series of instantaneous PM MN subspaces covering

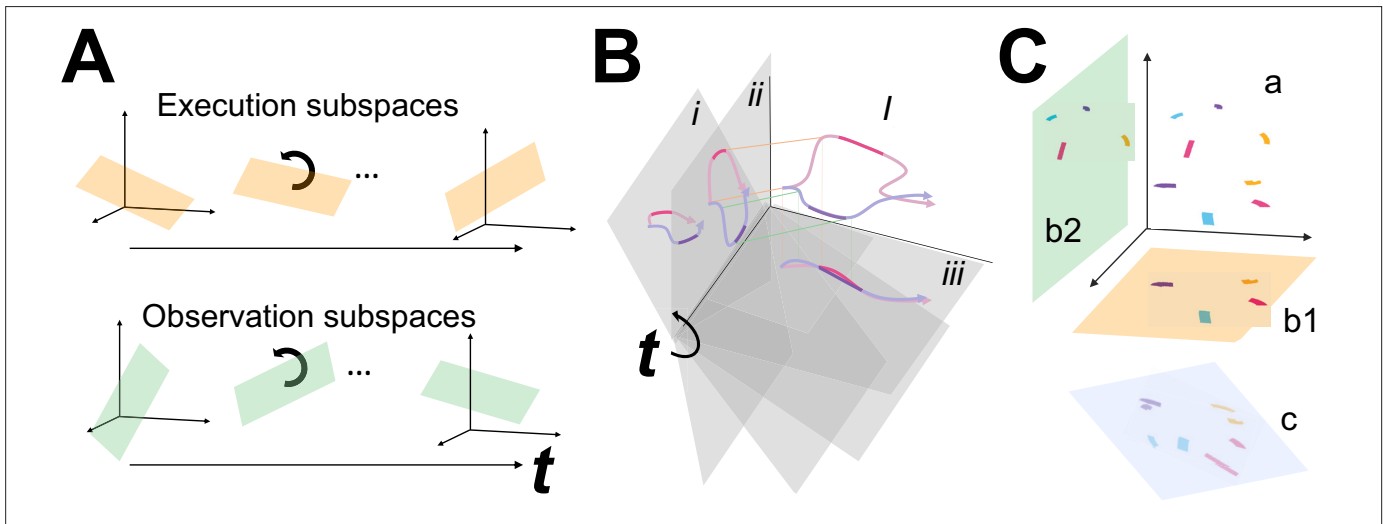

**Figure 1.** Conceptual approach. (**A**) We hypothesized that the condition-dependent instantaneous subspace of premotor cortex mirror neuron (PM MN) activity shifts progressively throughout the time course of behavioral trials both during execution (orange) and during observation (green). Such shifting can be examined by calculating the principal angles between a selected instantaneous subspace and every other subspace in the time series, *t*. (**B**) Segments clipped from the neural trajectories of two different movements (magenta, purple) in a high-dimensional space, *I*, show varying distance between them when projected into a time series (*t=i, ii, iii*) of shifting, low-dimensional instantaneous subspaces (gray). This varying distance indicative of the progressive shifting of the instantaneous subspace can be followed by decoding the different movements from the trajectory segments projected into the time series of instantaneous subspaces. (**C**) Neural trajectory segments from the four reach-grasp-manipulate (RGM) movements (magenta, purple, cyan, and yellow) during execution and during observation originate in the same high-dimensional space (a), but project into distinct low-dimensional execution (orange, b1) and observation (green, b2) subspaces. Nevertheless, canonical correlation analysis (CCA) may identify another subspace (pale blue, c) where the projected magenta, purple, cyan, and yellow segments from both execution and observation show a similar spatial relationship to one another, with the two segments of each color projecting close to one another. Such correlation between the two sets of trajectory segments projected into the same subspace would indicate similar latent dynamic relationships among the four movements during execution and observation.

the time course of behavioral trials. We identified separate time series for execution trials and for observation trials, both involving four different reach-grasp-manipulate (RGM) movements. Given that each subspace in these time series is instantaneous (a snapshot in time), it captures condition-dependent variance in the neural activity among the four RGM movements while minimizing condition-independent (time-dependent) variance.

We then tested the hypothesis that the condition-dependent subspace shifts progressively over the time course of behavioral trials (*Figure 1A*) by calculating the principal angles between four selected instantaneous subspaces that occurred at times easily defined in each behavioral trial—instruction onset (I), go cue (G), movement onset (M), and the beginning of the final hold (H)—and every other instantaneous subspace in the time series. Initial analyses showed that condition-dependent neural trajectories for the four RGM movements tended to separate increasingly over the course of behavioral trials. We therefore additionally examined the combined effects of (i) the progressively shifting subspaces and (ii) the increasing trajectory separation, by decoding neural trajectory segments sampled for 100 ms after times I, G, M, and H and projected into the time series of instantaneous subspaces (*Figure 1B*).

Finally, we used canonical correlation to ask whether the prevalent patterns of MN co-modulation showed similar relationships among the four RGM movements during execution and observation (*Figure 1C*). Such alignment would indicate that the relationships among the trajectory segments in the execution subspace are similar to the relationships among the trajectory segments in the observation subspace, indicating a corresponding structure in the latent dynamic representations of execution and observation movements by the same PM MN population. And finally, because we previously have found that during action execution, the activity of PM MNs tends to lead that of non-MNs which are active only during action execution (AE neurons) (*Mazurek and Schieber, 2019*), we performed parallel analyses of the instantaneous state space of PM AE neurons.

## Results

We recorded spiking activity as each of three monkeys executed a delayed response RGM task, and then as each monkey observed the same task being performed by an experimenter (*Figure 2A*). Because we chose to study relatively naturalistic movements, the RGM components were not performed separately, but rather in a continuous fluid motion during the movement epoch of the task

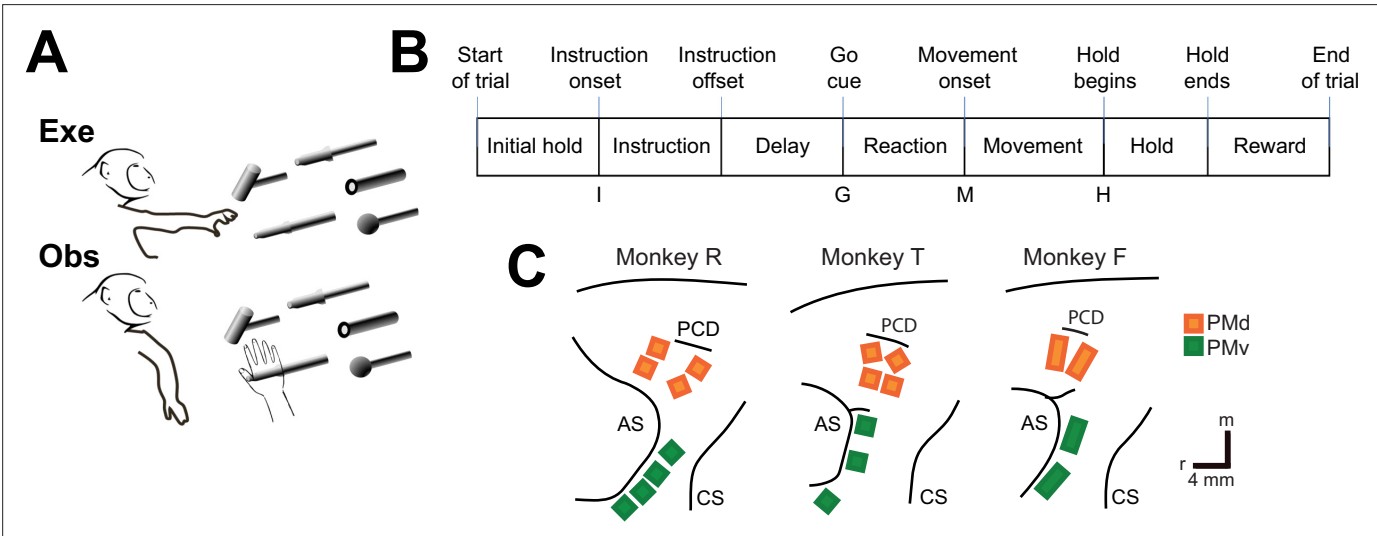

**Figure 2.** The reach-grasp-manipulate (RGM) task. (**A**) In separate blocks of trials, monkeys reached for, grasped, and manipulated four different objects themselves (Exe) and then observed a human performing the same task (Obs). (**B**) The times of eight behavioral events from start-of-trial to end-of-trial divided each trial into seven epochs from initial hold to reward. For analyses, the data were aligned separately on, and trajectories were sampled for 100 ms following the times of four selected events—instruction onset (I), go cue (G), movement onset (M), and the beginning of the final hold (H). (**C**) Recording array locations in ventral premotor cortex (PMv) (green) and dorsal premotor cortex (PMd) (orange) for each monkey have been redrawn from intraoperative photographs. PCD—precentral dimple; AS—arcuate sulcus; CS—central sulcus; r—rostral; m—medial. Scale bars representing 4 mm apply to all three monkeys.

**Table 1.** Numbers of trials in each session.
For each of the three sessions from each of the three monkeys, numbers of trials involving each of the four objects (sphere, button, coaxial cylinder, perpendicular cylinder) are given in parentheses separately for execution and for observation.

|  | Monkey R | | Monkey F | | Monkey T | |
|---|---|---|---|---|---|---|
|  | Exe | Obs | Exe | Obs | Exe | Obs |
| Session 1 | (22,8,25,26) | (32,31,30,31) | (58,59,62,63) | (71,72,71,72) | (57,54,57,55) | (60,61,59,57) |
| Session 2 | (34,26,34,38) | (40,41,40,37) | (59,58,60,56) | (73,72,75,74) | (47,53,52,43) | (57,53,58,58) |
| Session 3 | (42,41,49,45) | (49,50,51,49) | (63,58,58,58) | (72,75,74,74) | (43,41,38,42) | (50,48,48,50) |

sequence (*Figure 2B*). In previous studies involving a version of this task without separate instruction and delay epochs, we have shown that joint kinematics, EMG activity, and neuron activity in the M1 all vary throughout the movement epoch in relation to both reach location and object grasped, with location predominating early in the movement epoch and object predominating later (*Rouse and Schieber, 2015*; *Rouse and Schieber, 2016a*; *Rouse and Schieber, 2016b*). The present task, however, did not dissociate the reach, the hand shape used to grasp the object, and the manipulation performed on the object. Additional details of the behavioral task are described in the Methods. Three sessions were recorded from each of the three monkeys, R, F, and T (a 6 kg female, 10 kg male, and 10 kg male, respectively). The numbers of successful execution trials (Exe) and observation trials (Obs) involving each of the four objects—sphere, button, coaxial cylinder, and perpendicular cylinder—are given in *Table 1*.

The three monkeys each were implanted with floating microelectrode arrays (FMAs, Microprobes for Life Sciences) in the PMv and in the PMd. The locations of the arrays in each monkey are illustrated in *Figure 2C*. Using object and epoch as factors, we performed two-way repeated measures analysis of variance (ANOVA) on the firing rate of each sorted unit recorded from the arrays in each session (see Methods). Because unit firing rates typically differed during execution and observation, we performed such ANOVAs separately on execution trials and observation trials. *Table 2* gives the numbers of PM (PMv+PMd) units identified in each session as being modulated significantly during both execution and observation, which we refer to as MNs, along with the numbers of units modulated significantly during execution but not observation (AE), during observation but not execution (AO), or with no significant modulation during either execution or observation (NS). The numbers of

**Table 2.** Numbers of premotor cortex units in each session.
For each of the three sessions from each of the three monkeys (R, T, and F), numbers of PM units are given for each of four classes in the format of total (PMv, PMd). MNs—mirror neurons, modulated significantly during action execution and during action observation. AE—action execution neurons, modulated during execution but not during observation. AO—action observation neurons, modulated during observation but not execution. NS—not significant, units not modulated significantly during either execution or observation.

| Monkey | Session | MN | AE | AO | NS |
|---|---|---|---|---|---|
|  | 1 | 48 (19,29) | 35 (20,15) | 3 (1,2) | 5 (2,3) |
|  | 2 | 47 (21,26) | 25 (16,9) | 5 (1,4) | 11 (4,7) |
| R | 3 | 37 (19,18) | 49 (20,29) | 1 (1,0) | 8 (7,1) |
|  | 1 | 79 (37,42) | 15 (5,10) | 2 (0,2) | 7 (1,6) |
|  | 2 | 91 (48,43) | 22 (6,16) | 3 (1,2) | 7 (1,6) |
| T | 3 | 100 (48,52) | 18 (7,11) | 0 (0,0) | 6 (2,4) |
|  | 1 | 44 (24,20) | 7 (5,2) | 1 (1,0) | 8 (8,0) |
|  | 2 | 47 (32,15) | 10 (9,1) | 5 (1,4) | 3 (3,0) |
| F | 3 | 42 (28,14) | 9 (7,2) | 3 (1,2) | 3 (3,0) |

AO and NS units were consistently small across monkeys and sessions. The present analyses therefore focus on MNs and, for comparison, AE neurons.

## Condition-dependent versus condition-independent neural activity in PM MNs

Whereas a large fraction of condition-dependent neural variance during reaching movements without grasping can be captured in a two-dimensional subspace (*Churchland et al., 2012*; *Ames et al., 2014*), condition-dependent activity in movements that involve grasping is more complex (*Suresh et al., 2020*). In part, this may reflect the greater complexity of controlling the 24 degrees of freedom (DOFs) in the hand and wrist as compared to the 4 DOFs in the elbow and shoulder (*Sobinov and Bensmaia, 2021*). *Figure 3* illustrates this complexity in a PM MN population during the present RGM movements. Here, PCA was performed on the activity of a PM MN population across the entire time course of execution trials involving all four objects. The colored traces in *Figure 3A* show neural trajectories averaged separately across trials involving each of the four objects and then projected into the PC1 versus PC2 plane of the total neural space. Most of the variance in these four trajectories is comprised of a shared rotational component. The black trajectory, obtained by averaging trajectories from trials involving all four objects together, represents this condition-independent (i.e. independent of the object involved) activity. The condition-dependent (i.e. dependent on which object was involved) variation in activity is reflected by the variation in the colored trajectories around the black trajectory. The condition-dependent portions can be isolated by subtracting the black trajectory from each of the colored trajectories. The resulting four condition-dependent trajectories have been projected into the PC1 versus PC2 plane of their own common subspace in *Figure 3B*. Rather than exhibiting a simple rotational motif, these trajectories appear knotted. To better understand how these complex, condition-dependent trajectories progress over the time course of RGM trials, we chose to examine time series of instantaneous subspaces.

## Instantaneous subspaces shift progressively during both execution and observation

We identified an instantaneous subspace at each 1 ms time step of RGM trials. At each time step, we applied PCA to the 4 instantaneous neural states (i.e. the 4 points on the neural trajectories representing trials involving the 4 different objects each averaged across 20 trials per object, totaling 80 trials), yielding a three-dimensional subspace at that time (see Methods). Note that because these three-dimensional subspaces are essentially instantaneous, they capture the condition-dependent variation in neural states, but not the common, condition-independent variation. To examine the temporal progression of these instantaneous subspaces, we then calculated the principal angles between each 80-trial instantaneous subspace and the instantaneous subspaces averaged across *all* trials at four behavioral time points that could be readily defined across trials, sessions, and monkeys: the onset of the instruction (I), the go cue (G), the movement onset (M), and the beginning of the final hold (H). This process was repeated 10 times with replacement to assess the variability of the principal angles. The closer the principal angles are to 0°, the closer the two subspaces are to being identical; the closer to 90°, the closer the two subspaces are to being orthogonal.

Figure 4A–D illustrates the temporal progression of the first principal angle of the MN population in the three sessions (red, green, and blue) from monkey R during execution trials. As illustrated in *Figure 4—figure supplement 1* (see also the related Methods), in each session, all three principal angles, each of which could range from 0° to 90°, tended to follow a similar time course. In the Results, we therefore illustrate only the first (i.e. smallest) principal angle. Solid traces represent the mean across 10-fold cross-validation using the 80-trial subsets of all the available trials; shading indicates ±1 standard deviation. As would be expected, the instantaneous subspace using 80 trials approaches the subspace using all trials at each of the four selected times—I, G, M, and H—indicated by the relatively narrow trough dipping toward 0°. Of greater interest are the slower changes in the first principal angle in between these four time points. *Figure 4A* shows that after instruction onset (I), the instantaneous subspace shifted quickly away from the subspace at time I, indicated by a rapid increase in principal angle to levels not much lower than what might be expected by chance alone (horizontal dashed line). In contrast, throughout the remainder of the instruction and delay epochs (from I to G), *Figure 4B and C* show that the 80-trial instantaneous subspace shifted gradually and concurrently, not sequentially,

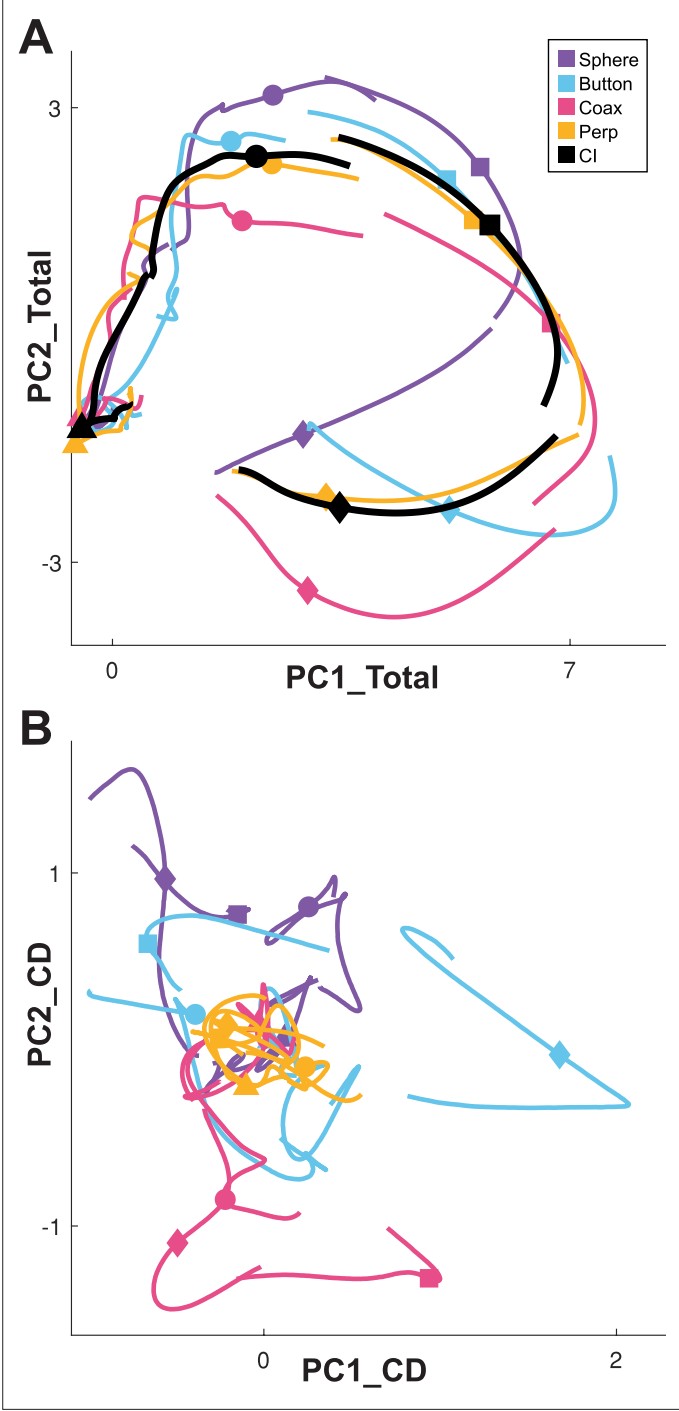

**Figure 3.** Neural trajectories of condition-independent versus condition-dependent activity. (**A**) Neural trajectories of premotor cortex mirror neuron firing rates averaged across multiple execution trials involving each of the four objects (sphere—purple, button—cyan, coaxial cylinder [Coax]—magenta, perpendicular cylinder [Perp]—yellow) have been projected into the PC1 versus PC2 plane of the total neural activity. Averaging these four trajectories gives their common, condition-independent (CI) trajectory (black). Time proceeds clockwise from left, with data separately aligned at four selected times: triangle—instruction onset (I); circle—go cue (G); square—movement onset (M); diamond—beginning of final hold (H). (**B**) Condition-dependent trajectories obtained by subtracting the CI trajectory (black) from each of the four single-object trajectories (colors) in (**A**), and then projected into the PC1 versus PC2 plane of their common, condition-dependent (CD) subspace across the entire time course of trials. Data from monkey R, session 2.

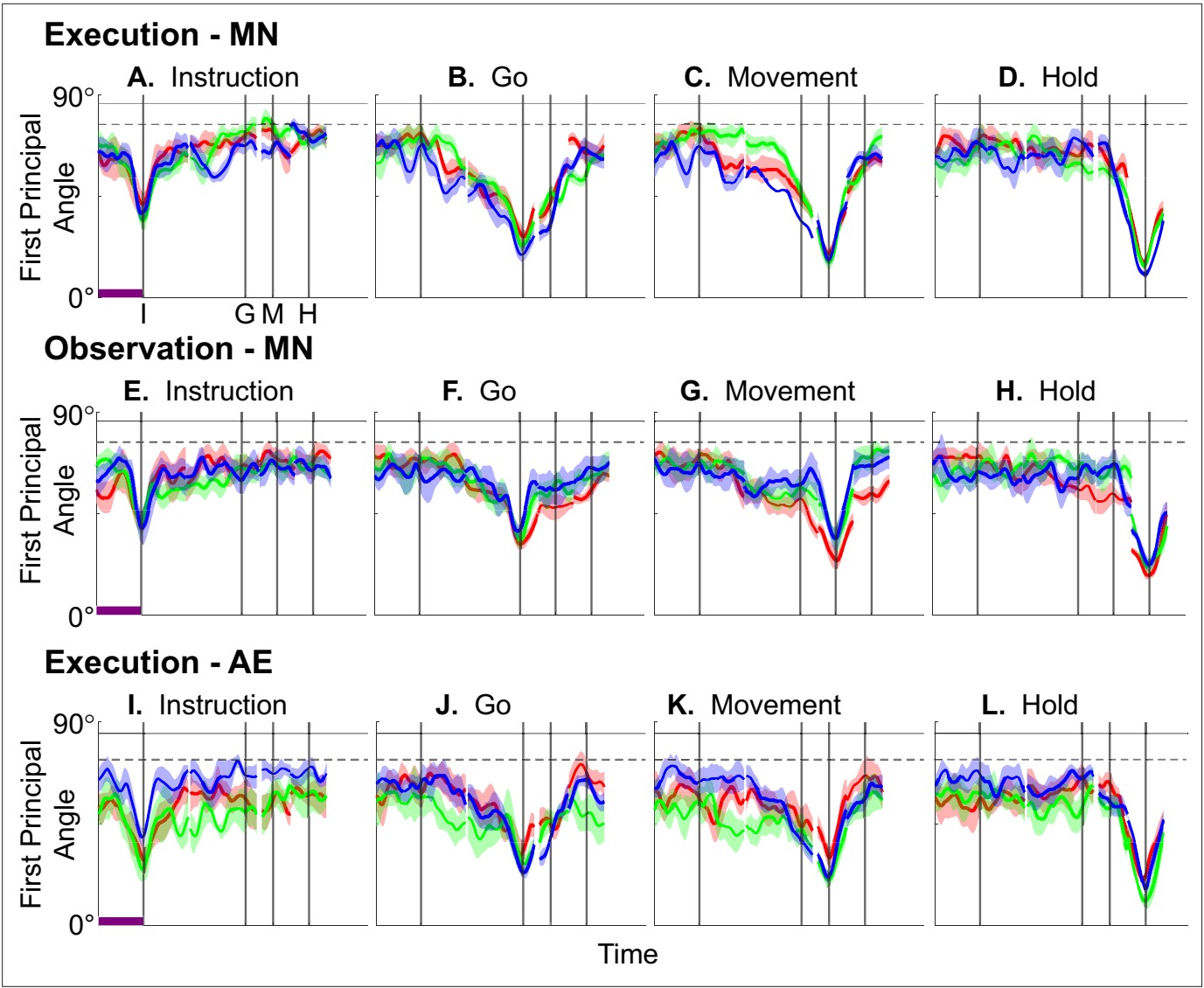

**Figure 4.** Time course of the first principal angle between instantaneous subspaces. (**A–D**) Mirror neuron (MN) populations during execution trials; (**E–H**) MN populations during observation trials; (**I–L**) AE neuron populations during execution trials. Each frame shows the time course of the first principal angle between the time series of instantaneous subspaces and that present at one of four selected times—A, E, I: instruction onset; B, F, J: go cue; C, G, K: movement onset; or D, H, L: the beginning of the final hold. Results in 1 ms steps have been aligned separately at the times of the instruction onset (**I**), go cue (**G**), movement onset (**M**), and hold (**H**) —each indicated by a vertical line as labeled in the frame at upper left. Red, green, and blue traces represent sessions 1, 2, and 3, respectively, from monkey R. Solid traces represent means, and shaded areas represent ±1 standard deviation across 10-fold cross-validation as described in the Methods. Horizontal black lines indicate the average (solid) and the average minus 3 standard deviations (dashed) of the first principal angle between a fixed 3D space and other 3D spaces chosen randomly within an *N*-dimensional space (see *Figure 4— figure supplement 2* and related Methods). Here, *N*=37, the number of MNs in session 3. Horizontal purple bars in the left column (**A, E, I**) indicate 500 ms, which applies to the entire row.

The online version of this article includes the following figure supplement(s) for figure 4:

**Figure supplement 1.** First, second, and third principal angles as a function of time.

**Figure supplement 2.** First principal angles between a fixed 3D subspace and 5000 other 3D subspaces randomly chosen from spaces of dimensionality, *N*, varying from 5 to 500.

**Figure supplement 3.** Time course of the first principal angle of instantaneous subspaces for AE neurons during observation trials.

toward the all-trial subspaces that would be reached at the end of the delay period (G) and then at the onset of movement (M), indicated by the progressive decreases in principal angle. As shown by *Figure 4D*, shifting toward the H subspace did not begin until the movement onset (M). To summarize, these changes in principal angles indicate that after shifting briefly toward the subspace present at time the instruction appeared (I), the instantaneous subspace shifted progressively throughout the instruction and delay epochs toward the subspace that would be reached at the time of the go cue (G), then further toward that at the time of movement onset (M), and only thereafter shifted toward the instantaneous subspace that would be present at the time of the hold (H).

*Figure 4E–H* shows the progression of the first principal angle of the MN population during observation trials. Overall, the temporal progression of the MN instantaneous subspace during observation was similar to that found during execution, particularly around times I and H. The decrease in principal angle relative to the G and M instantaneous subspaces during the delay epoch was less pronounced during observation than during execution. Nevertheless, these findings support the hypothesis that the condition-dependent subspace of PM MNs shifts progressively over the time course of RGM trials during both execution and observation, as illustrated schematically in *Figure 1A*.

We also examined the temporal progression of the instantaneous subspace of AE neurons. As would be expected given that AE neurons were not modulated significantly during observation trials, in the observation context AE populations had no gradual changes in principal angle (*Figure 4—figure supplement 3*). During execution, however, *Figure 4I–L* shows that the AE populations had a pattern of gradual decrease in principal angle similar to that found in the MN population (*Figure 4A–D*). After the instruction onset, the instantaneous subspace shifted quickly away from that present at time I and progressed gradually toward that present at times G and M, only shifting toward that present at time H after movement onset. As for the PM MN populations, the condition-dependent subspace of the PM AE populations shifted progressively over the time course of execution RGM trials.

## Neural trajectories separate progressively during both execution and observation

The progressive changes in principal angles do not capture another important aspect of condition-dependent neural activity. The neural trajectories during trials involving different objects separated increasingly as trials progressed in time. To illustrate this increasing separation, we clipped 100 ms segments of high-dimensional MN population trial-averaged trajectories beginning at times I, G, M, and H, for trials involving each of the four objects. We then projected the set of four object-specific trajectory segments clipped at each time into each of the four instantaneous 3D subspaces at times I, G, M, and H. This process was repeated separately for execution trials and for observation trials.

For visualization, we projected these trial-averaged trajectory segments from an example session into the PC1 versus PC2 planes (which consistently captured >70% of the variance) of the I, G, M, or H instantaneous 3D subspaces. In *Figure 5*, the trajectory segments for each of the four objects (sphere—purple, button—cyan, coaxial cylinder—magenta, perpendicular cylinder—yellow) sampled at different times (rows) have been projected into each of the four instantaneous subspaces defined at different times (columns). Rather than appearing knotted as in *Figure 3*, these short trajectory segments are distinct when projected into each instantaneous subspace.

Along the main diagonal of *Figure 5A*, each set of trajectory segments is projected into its corresponding subspace, showing that during execution the trajectory segments for the four objects were close together at the time of instruction onset (I), became more separated at the time of the go cue (G), had separated further still at movement onset (M), and had become somewhat less separated at the beginning of the final hold (H). During observation (*Figure 5B*), a similar trend is evident along the main diagonal, although the separation is less, reflecting the commonly described lower firing rates of MNs during observation than during execution (*Ferroni et al., 2021*). In addition, during observation, the separation of the four trajectories was somewhat greater at the beginning of the hold (H) than at movement onset (M). Off-diagonal frames along the rows (same trajectory segments, different instantaneous subspaces) or along the columns (different trajectory segments, same instantaneous subspaces) show less separation than along the main diagonal, both during execution and during observation. To summarize these differences in trajectory separation, we calculated the three-dimensional cumulative separation (*CS*—see Methods) for each set of four segments projected into each of the four instantaneous subspaces both for this example session and averaged across all nine

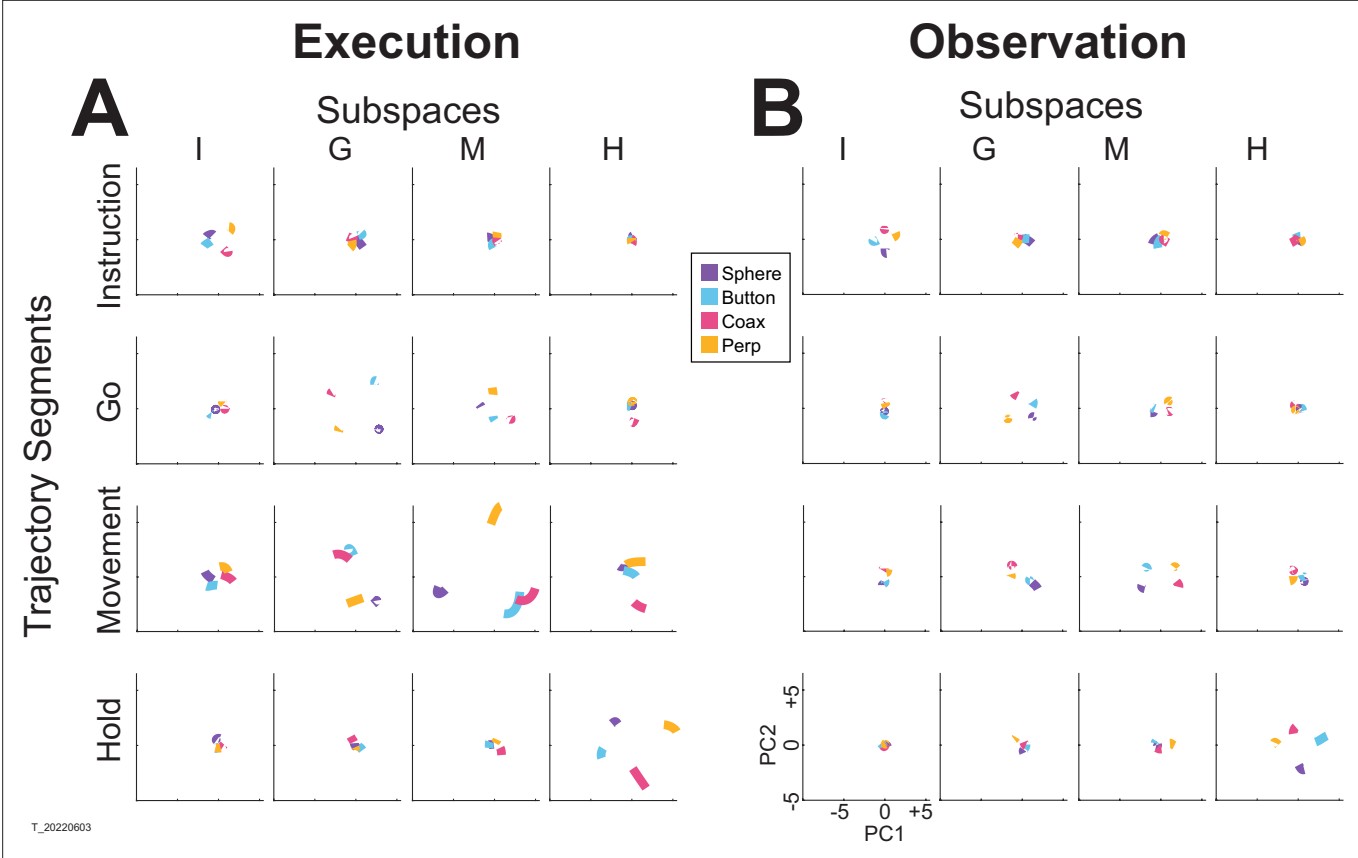

**Figure 5.** Mirror neuron trajectory segments projected into instantaneous subspaces. (**A**) Using execution data from an example session (monkey T, session 3), trajectory segments averaged across trials involving each of the four objects (sphere—purple, button—cyan, coaxial cylinder [coax]—magenta, perpendicular cylinder [perp]—yellow) were clipped for 100 ms immediately following each of four behavioral events (rows: instruction onset, go cue, movement onset, hold). Each set of these four segments was then projected into the PC1 versus PC2 plane of the instantaneous 3D subspace present at four different times (columns: **I**, **G**, **M**, **H**). (**B**) The same process was performed using observation data from the same session. The PC1 versus PC2 scales at lower left in (**B**) apply to all frames in both (**A**) and (**B**).

The online version of this article includes the following figure supplement(s) for figure 5:

**Figure supplement 1.** Cumulative separation.

sessions (*Figure 5—figure supplement 1*). These differences in separation when the same trajectory segments are projected into different subspaces reflect the progressive shifting of the condition-dependent instantaneous subspace of the PM MN population as trials progressed over time, as illustrated schematically in *Figure 1B*.

## Decodable information changes progressively during both execution and observation

As RGM trials proceeded in time, the condition-dependent neural activity of the PM MN population thus changed in two ways. First, the instantaneous condition-dependent subspace shifted, indicating that the patterns of firing rate co-modulation among neurons representing the four different RGM movements changed progressively, both during execution and during observation. Second, as firing rates generally increased, the neural trajectories representing the four RGM movements became progressively more separated, more so during execution than during observation.

To evaluate the combined effects of these two progressive changes, we clipped 100 ms single-trial trajectory segments beginning at times I, G, M, or H, and projected these trajectory segments from individual trials into the instantaneous 3D subspaces at 50 ms time steps. At each of these time steps, we trained a separate long short-term memory (LSTM) decoder to classify individual trials according to which of the four objects was involved in that trial. We expected that the trajectory segments would

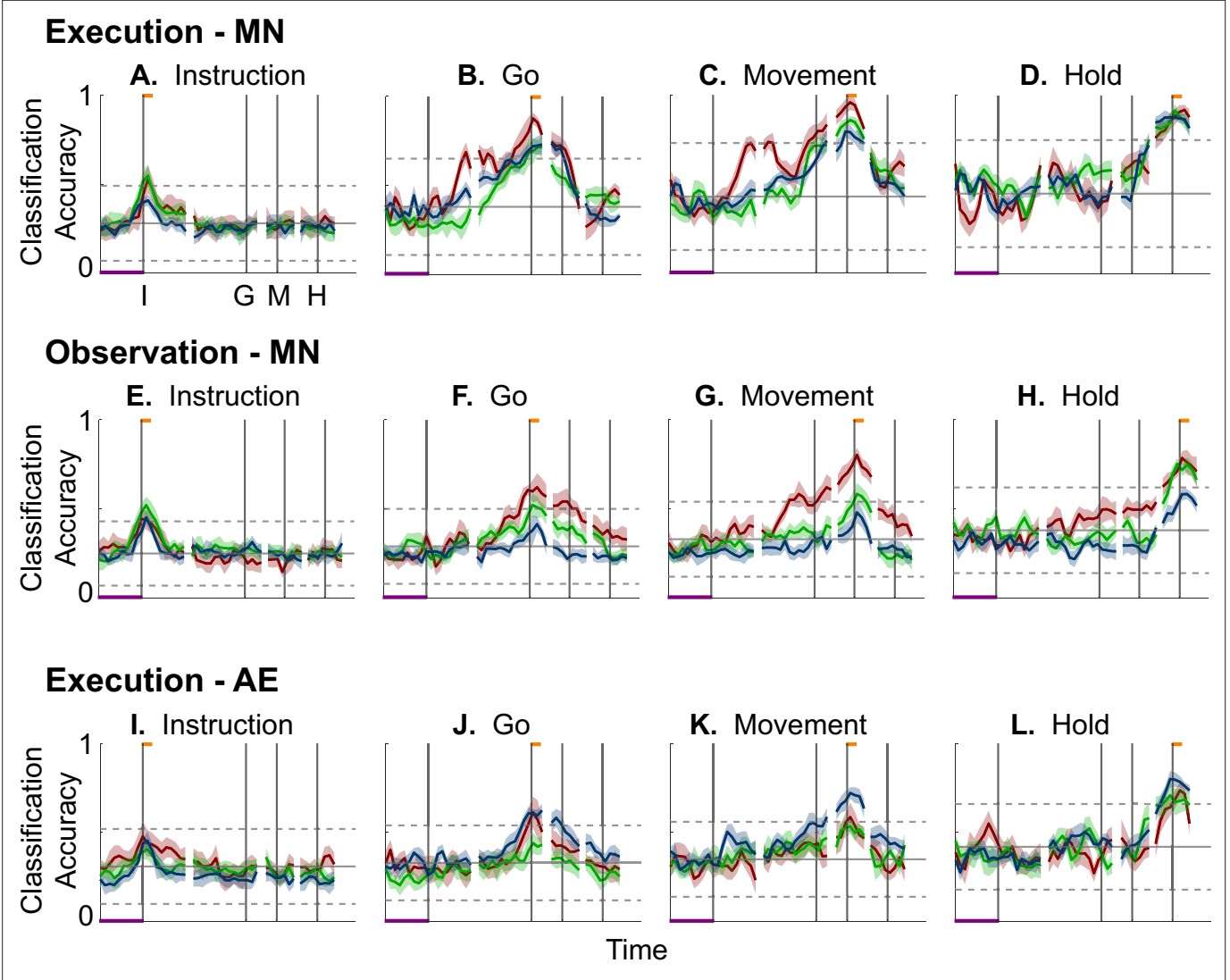

**Figure 6.** Decodable information as a function of time. (**A–D**) Classification accuracy for mirror neuron execution trajectory segments projected into instantaneous execution subspaces; (**E–H**) for mirror neuron observation trajectory segments projected into their instantaneous observation subspaces; (**I–L**) for action-execution neuron trajectory segment projected into their instantaneous execution subspaces. (**A, E, I**) Instruction trajectory segments; (**B, F, J**) go segments; (**C, G, K**) movement segments; (**D, H, L**) hold segments. Red, green, and blue traces represent sessions 1, 2, and 3, respectively, from monkey R. Results in 50 ms steps have been aligned separately at the times of the instruction onset (I), go cue (G), movement onset (M), and hold (H)—each indicated by a vertical line as labeled in the frame at upper left. In each frame, the short horizontal orange flag at the top of the vertical lines indicates the 100 ms during which each set of trajectory segments was clipped; the horizontal purple bar at lower left represents 500 ms. Solid curves indicate mean classification accuracy across 10-fold cross-validation as a function of time, with the shaded areas indicating 1 standard deviation. Horizontal black lines indicate the mean (solid) ±3 standard deviations (dashed) classification accuracy obtained by projecting each set of trajectory segments into 500 randomly selected 3D spaces.

The online version of this article includes the following figure supplement(s) for figure 6:

**Figure supplement 1.** Decodable information as a function of time in premotor cortex (PM) AE neuron populations.

be classified most accurately when projected into instantaneous subspaces near the time at which the trajectory segments were clipped. At other times, we reasoned that classification accuracy would depend both on the similarity of the current instantaneous subspace to that found at the clip time as evaluated by the principal angle (*Figure 4*) and on the separation of the four trajectories at the clip time (*Figure 5*).

*Figure 6A–D* shows the resulting classification accuracy as a function of trial time for the 100 ms instruction, go, movement, or hold MN execution trajectory segments, each projected into the same

time series of instantaneous MN execution subspaces from the same session. Solid curves indicate classification accuracy averaged across 10-fold cross-validation (as described in the Methods); the surrounding shaded areas indicate ±1 standard deviation from that average; different colors indicate results from the three different sessions in monkey R. Horizontal lines indicate the range of classification accuracies that would have been obtained had the instantaneous subspaces been chosen randomly, which we estimated for each set of trajectory segments by bootstrapping—projecting the trajectory segments into a randomly selected 3D space, training an LSTM decoder, and classifying single trials, repeated 500 times (*Natraj et al., 2022*).

As might have been expected based both on principal angles and on trajectory separation, classification accuracy consistently peaked at a time point within or near the 100 ms duration of the corresponding trajectory segments (orange flags at the top of the vertical lines). Classification accuracy decreased progressively at times preceding and following each of these peaks. In monkey R, mean classification of the instruction trajectory segments (*Figure 6A*) initially was ~0.25, rose toward ~0.50 around the time of the instruction onset, and then fell back to ~0.25. Mean accuracy for the go segments (*Figure 6B*) also began at ~0.25, rose gradually during the delay epoch to peak at ~0.75 around the time of the go cue, and decreased thereafter. For the movement (*Figure 6C*) and hold (*Figure 6D*) segments, classification accuracy started somewhat higher (reflecting greater trajectory segment separation at the time they were clipped, *Figure 5*) and peaked at ~0.90. Similar trends were seen for monkeys T and F. For each monkey, classification accuracy for each of the four sets of trajectory segments—instruction, go, movement, and hold—as a function of time was relatively consistent across sessions.

Although classification accuracy consistently peaked near the behavioral event at which time each set of trajectory segments was clipped, the rise in accuracy before and the decline after the peak differed depending on the behavioral event. Peak classification accuracy for instruction segments was modest, beginning to rise from mean chance levels ~100 ms before the instruction onset and quickly falling back thereafter (*Figure 6A*). At times outside of this brief peak, however, the instantaneous subspace was no more similar to that at the time of instruction onset than would be expected from chance alone.

In contrast, classification accuracy for the go trajectory segments (*Figure 6B*) was elevated above mean chance levels for more of the RGM trial duration. Though exceeding 3 standard deviations from mean chance only late in the delay epoch, go segment classification accuracy rose steadily through the delay epoch, peaked near the go cue, then fell back to near mean chance levels during the reaction (G to M) and movement (M to H) epochs. Likewise, the rise in classification accuracy of movement trajectory segments (*Figure 6C*) also began, not after the go cue, but earlier, in the middle of the delay epoch (I to G). Movement segment classification accuracy rose steadily from the second half of the delay epoch through the reaction epoch (G to M), peaked above chance levels shortly after movement onset (M), and fell back to near baseline during the movement epoch (M to H). Had the condition-dependent instantaneous subspaces during the delay epoch been orthogonal to those at the time of movement onset, the movement trajectory segments would have had no projection in delay epoch subspaces and classification accuracy would have remained at baseline. The progressive increase in classification accuracy of movement trajectory segments during the preparatory delay and reaction epochs indicates that as these epochs proceeded, the condition-dependent neural trajectories of PM MNs shifted gradually, not abruptly, toward where they would be at movement onset.

Classification accuracy of the hold trajectory segments (*Figure 6D*) increased relatively late in execution trials. During the instruction, delay, and reaction epochs, the instantaneous subspaces were no more similar than chance to that at the beginning of the hold epoch. Classification accuracy of hold trajectory segments began to increase only after movement onset (M), rising through the movement epoch, peaking near the beginning of the hold epoch and decreasing thereafter.

We performed a similar classification accuracy analysis for observation trials. For instruction trajectory segments (*Figure 6E*), the brief peak of classification accuracy occurring around the time of instruction onset (I) during observation trials was quite like that found during execution trials. For the go and movement segments (*Figure 6F and G*), although classification accuracy tended to be lower, a gradual rise again began during the delay epoch. Classification accuracy of the hold trajectory segments, during observation as during execution, began to increase only after movement onset (*Figure 6H*).

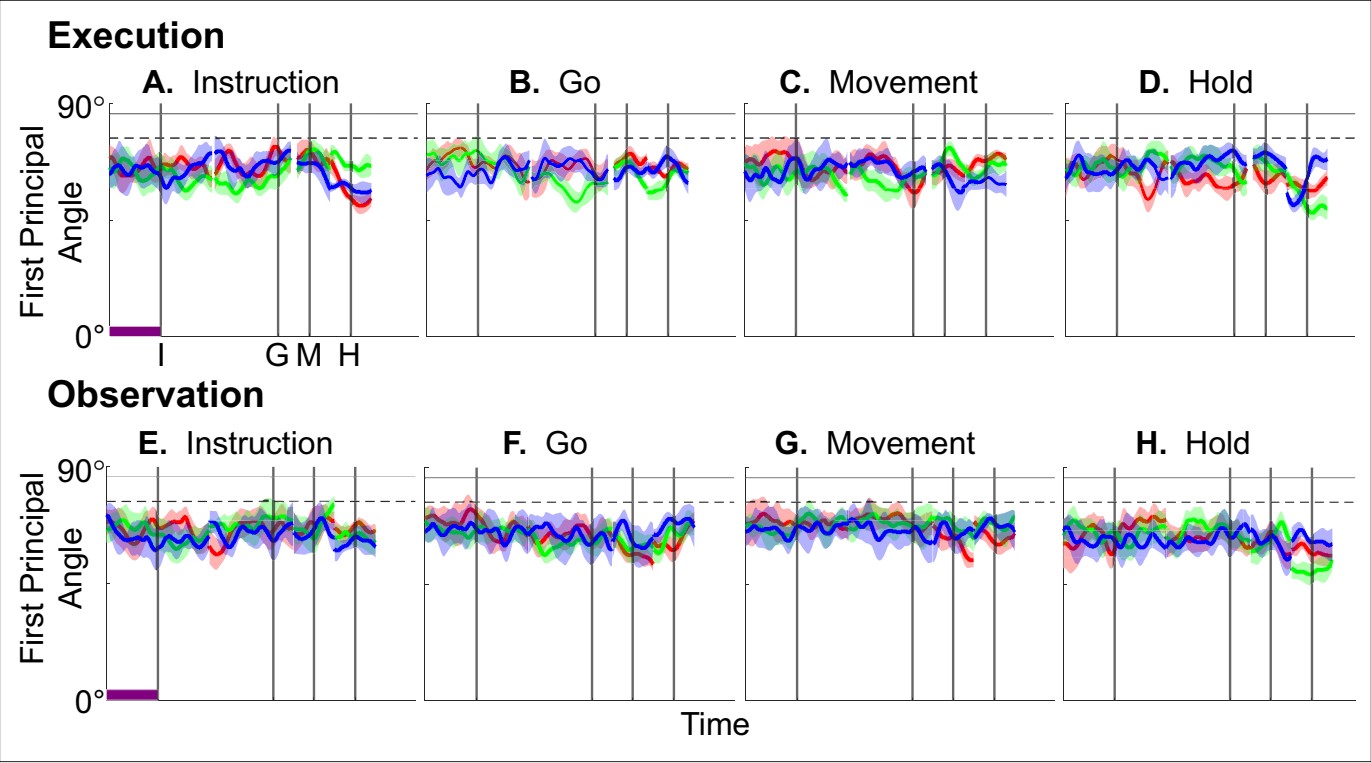

**Figure 7.** Time course of the first principal angle cross-calculated between instantaneous execution and observation subspaces of premotor cortex mirror neurons as a function of time. First principal angles between the instantaneous *execution* subspace at selected times I, G, M, or H and the entire time series of instantaneous *observation* subspaces are shown above (**A–D**); between the instantaneous *observation* subspace at selected times I, G, M, or H and the entire time series of instantaneous *execution* subspaces below (**E–H**). Formatting is the same as in *Figure 4*.

The online version of this article includes the following figure supplement(s) for figure 7:

**Figure supplement 1.** Classification accuracy of trajectory segments cross-projected between instantaneous execution and observation subspaces of premotor cortex mirror neurons as a function of time.

**Figure supplement 2.** Partial overlap of execution and observation subspaces in monkey T.

During execution trials, classification accuracy for AE populations (*Figure 6I–L*) showed a time course quite similar to that for MN populations, though amplitudes were lower overall, most likely because of the smaller population sizes. During observation, AE populations showed only low-amplitude, short-lived peaks of classification accuracy around times I, G, M, and H (*Figure 6—figure supplement 1*). Given that individual AE neurons showed no statistically significant modulation during observation trials, even these small peaks might not have been expected. Previous studies have indicated, however, that neurons not individually related to task events nevertheless may contribute to a population response (*Shenoy et al., 2013*; *Cunningham and Yu, 2014*; *Gallego et al., 2017*; *Jiang et al., 2020*).

## Do PM MNs progress through the same subspaces during execution and observation?

Having found that PM MN populations show similar progressive shifts in their instantaneous neural subspace during execution and observation of RGM trials, as well as similar changes in decodable information, we then asked whether this progression passes through similar subspaces during execution and observation. To address this question, we first calculated the principal angles between the instantaneous MN *execution* subspace at selected times I, G, M, or H and the entire time series of instantaneous MN *observation* subspaces (*Figure 7A–D*). Conversely, we calculated the principal angles between the instantaneous *observation* subspaces at selected times I, G, M, or H and the entire time series of instantaneous *execution* subspaces (*Figure 7E–H*). Although the principal angles were slightly smaller than might be expected from chance alone, indicating some minimal overlap of

execution and observation instantaneous subspaces, the instantaneous observation subspaces did not show any progressive shift toward the I, G, M, or H execution subspace (*Figure 7A–D*), nor did the instantaneous execution subspaces shift toward the I, G, M, or H observation subspace (*Figure 7E–H*). We also used classification accuracy to evaluate cross-projected trajectory segments and again found little evidence of overlap between execution and observation subspaces (*Figure 7—figure supplement 1*). Although monkey T did show evidence of some degree of overlap (*Figure 7—figure supplement 2*), throughout the time course of trials in monkeys R and F, the instantaneous execution and observation condition-dependent subspaces showed little, if any, overlap.

## Alignment of latent dynamics

We next asked whether MN execution and observation trajectory segments, though progressing through distinct subspaces, nevertheless could be aligned using canonical correlation analysis (CCA) to project both sets of trajectory segments into another, common subspace, as illustrated schematically in *Figure 1C*. Such alignment would indicate that neural representations of trials involving the four objects bore a similar relationship to one another in neural space during execution and observation, even though they occurred in different subspaces. For example, the trajectories of PMd+M1 neuron populations recorded from two different monkeys during center-out reaching movements could be aligned well (*Safaie et al., 2023*). CCA showed, for example, that in both brains, the neural trajectory for the movement to the target at 0° was closer to the trajectory for movement to the target at 45° than to the trajectory for the movement to the target at 180°. Relationships among these latent dynamic representations of the eight movements were thus similar even though the neural populations were recorded from two different monkeys.

We therefore applied CCA (see Methods) to align the trajectory segments of execution trials with those of observation trials. As an example, trial-averaged hold execution trajectory segments in their original execution subspace at time H and hold observation trajectory segments in their original observation subspace at time H are shown in *Figure 8A*. The relationships among the execution trajectory segments appear substantially different than that among the observation trajectory segments. But when both sets of trajectory segments are projected into another common subspace identified with CCA, as shown in *Figure 8B*, a similar relationship among the neural representations of the four movements during execution and observation is revealed. In both behavioral contexts, the neural representation of movements involving the sphere (purple) is now closest to the representation of movements involving the coaxial cylinder (magenta) and farthest from that of movements involving the button (cyan). The two sets of trajectory segments are more or less 'aligned'.

As a positive control, we first aligned MN execution trajectory segments from two different sessions in the same monkey (which we abbreviate as MN:1/2). The two sessions in monkey R provided only one possible comparison, but the three sessions in monkeys T and F each provided three comparisons. For each of these seven comparisons, we found the bootstrapped average of CC1, of CC2, and of CC3. The 3D means ± standard deviations of these seven averages for the instruction, go, movement, and hold trajectory segments have been plotted in *Figure 8C* (black). The progressive increase in mean correlation coefficients (CCs) reflects the general increase in firing rates relative to trial-by-trial variability from the early to later trial epochs. The highest values for MN:1/2 correlations were obtained for the movement trajectory segments ($\overline{CC1} = 0.89$, $\overline{CC2} = 0.77$, $\overline{CC3} = 0.61$). These relatively high values indicate relatively consistent relationships among the movement neural trajectory segments representing the four different RGM movements from session to session, as would have been expected from previous studies (*Gallego et al., 2018*; *Gallego et al., 2020*; *Safaie et al., 2023*).

Given that PM MN activity progressed largely through nonoverlapping instantaneous subspaces during execution versus observation, we proceeded to ask whether the relationship among the neural representations of the four RGM movements was similar during execution versus observation. To address this question, we aligned MN execution trajectory segments with MN observation trajectory segments from the same session (MN:E/O; two sessions from monkey R, three from monkey T, three from monkey F). The 3D mean ± standard deviation CCs for these eight alignments also have been plotted in *Figure 8C* (red). Here, the highest values were reached for the hold trajectory segments ($\overline{CC1} = 0.73$, $\overline{CC2} = 0.54$, $\overline{CC3} = 0.39$). Though not as high as for execution/execution alignment, these values indicate substantial alignment of MN trajectory segments from execution and observation. PM MN populations thus showed some degree of similarity in the relationships among

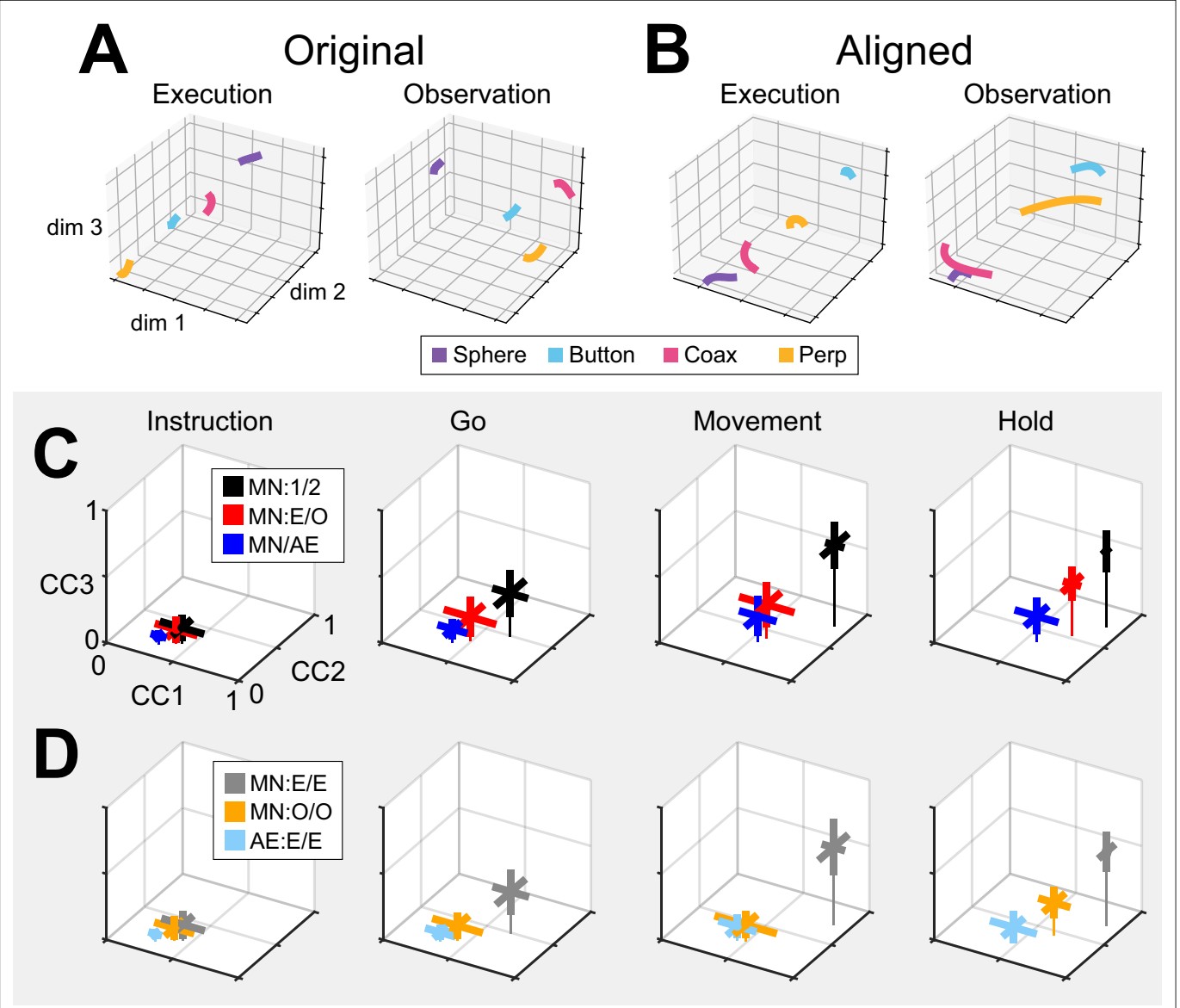

**Figure 8.** Alignment of trajectory segments by canonical correlation. (**A**) For an example session (monkey F, session 2), mirror neuron (MN) hold trajectory segments from execution trials have been projected into their original instantaneous execution subspace at time H (left) and from observation trials into their original instantaneous observation subspace also at time H (right). (**B**) The same execution (left) and observation (right) trajectory segments all have been projected into another, common subspace identified with canonical correlation. Colors indicate trajectory segments from trials involving the sphere—purple, coaxial cylinder (coax)—magenta, perpendicular cylinder (perp)—yellow, and button—cyan. (**C**) The three correlation coefficients resulting from canonical correlation analysis (CCA) (CC1, CC2, and CC3) have been averaged across comparisons from all sessions from the three monkeys. Thick bars representing the standard deviations of the three coefficients cross at their means, with a thin line dropped vertically from that point to the CC1 versus CC2 plane. CCA of MN trajectory segments from execution trials recorded in two different sessions from the same monkey (black, MN:1/2) is used as a point of reference with which to compare alignment of MN execution versus observation trials collected in the same session (red, MN:E/O) and MN versus AE neuron execution segments from the same session (blue, MN/AE). (**D**) Correlation coefficients from within-group CCA alignment for MN execution segments (gray, MN:E/E), MN observation trajectory segments (orange, MN:O/O), and AE execution segments (light blue, AE:E/E). See text for further description.

their latent dynamic representations of the four RGM movements during execution and observation, particularly at the time of the hold.

Although MNs are known to be present in considerable numbers in both the M1 and PM (see Introduction), most studies of movement-related cortical activity in these areas make no distinction between neurons with activity only during action execution (AE neurons) and those with activity during

both execution and observation (MNs). This reflects an underlying assumption that during action execution, MNs function in parallel with AE neurons, differing only during observation. We therefore tested the hypothesis that MN and AE neuron execution trajectory segments from the same session would align well. *Figure 8C* (blue) shows the mean CCs between MN and AE execution trajectory segments across eight alignments (MN/AE; 2R, 3T, 3F), which reached the highest values for the hold segments ($\overline{CC1} = 0.57$, $\overline{CC2} = 0.35$, $\overline{CC3} = 0.19$). All three of these coefficients were substantially lower than those for the MN execution versus observation alignments given above. Surprisingly, the alignment of AE neuron execution trajectory segments with those of the simultaneously recorded MN population was weaker than the alignment of MN trajectories during execution versus observation.

Statistical comparisons across the three sets of alignments illustrated in *Figure 8C* (MN:1/2; MN:E/O; and MN/AE) showed significant variation in each of the three CCA coefficients for each set of trajectory segments, with the exception of the instruction segments which were all quite low (Kruskal-Wallis tests; instruction segments, $p > 0.05$; go segments, $p < 0.01$; movement segments, $p < 0.01$; hold segments, $p < 0.001$). Post hoc testing showed that in all significant cases (9 cases: 3 CCA coefficients × 3 sets of trajectory segments, Tukey's honestly significant difference tests), though the MN:E/O coefficients might not be significantly lower than the corresponding MN/1:2 coefficients and/or significantly higher than the MN/AE coefficients, the MN/AE coefficients were significantly lower than the corresponding MN/1:2 coefficients in all 9 cases. These findings fail to support the hypothesis that during action execution, MN and AE neuron trajectory segments would align well and suggest instead that the patterns of co-modulation among AE neurons during the four different RGM movements did not align with the patterns of co-modulation among MNs.

Did these differences in MN:1/2, MN:E/O, and MN/AE alignment result from consistent differences in their respective patterns of co-modulation, or from greater trial-by-trial variability in the patterns of co-modulation among MNs during observation than during execution, and still greater variability among AE neurons during execution? The bootstrapping approach we used for CCA (see Methods) enabled us to evaluate the consistency of relationships among trajectory segments across repeated samplings of trials recorded from the same neuron population in the same session and in the same context (execution or observation). We therefore performed 500 iterations of CCA between two different random samples of MN execution (MN:E/E), MN observation (MN:O/O), or AE execution (AE:E/E) trajectory segments from a given session (2R, 3T, 3F). This within-group alignment of MN execution trajectory segments from the same session (*Figure 8D*, MN:E/E, gray, hold: $\overline{CC1} = 0.88$, $\overline{CC2} = 0.74$, $\overline{CC3} = 0.55$) was as strong as between-session alignment (*Figure 8C*, MN/1:2, black). But within-group alignment of MN observation trajectory segments (*Figure 8D*, MN:O/O, orange, hold: $\overline{CC1} = 0.65$, $\overline{CC2} = 0.46$, $\overline{CC3} = 0.24$) was lower than that found with MN execution segments (*Figure 8C*, MN:E/O, red, $\overline{CC1} = 0.73$, $\overline{CC2} = 0.54$, $\overline{CC3} = 0.39$). Likewise, within-group alignment of AE neuron trajectory segments (*Figure 8D*, AE:E/E, light blue, hold: $\overline{CC1} = 0.46$, $\overline{CC2} = 0.25$, $\overline{CC3} = 0.10$) was lower than their alignment with MN execution segments (*Figure 8C*, MN/AE, blue, hold: $\overline{CC1} = 0.57$, $\overline{CC2} = 0.35$, $\overline{CC3} = 0.19$). Whereas MN execution trajectories were relatively consistent within sessions, MN observation trajectories and AE execution trajectories were less so.

Statistical comparisons across these three sets of within-group alignments (MN:E/E; MN:O/O; and AE:E/E) showed significant variation in each of the three CCA coefficients for all four trajectory segments (Kruskal-Wallis tests; instruction segments, $p < 0.05$; go segments, $p < 0.01$; movement segments, $p < 0.001$; hold segments, $p < 0.001$). Post hoc testing showed that in all significant cases (12 cases: 3 CCA coefficients × 4 sets of trajectory segments, Tukey's honestly significant difference tests), though the within-group MN:O/O coefficients might not be significantly lower than the corresponding MN:E/E coefficients and/or significantly higher than the AE:E/E coefficients, the within-group AE:E/E coefficients were significantly lower than the corresponding MN:E/E coefficients in all 12 cases. These findings suggest that the patterns of co-modulation among AE neurons during the four different RGM movements, as well as the patterns of co-modulation among MNs during observation, were more variable from trial to trial than were the patterns of MN co-modulation during execution. This greater trial-to-trial variability in co-modulation of MNs during observation, and even greater variability in AE neurons during execution (*Figure 8D*), likely contribute to the weaker alignment of MN observation segments with MN execution segments and even weaker alignment of AE and MN execution segments (*Figure 8C*). Whereas the predominant patterns of co-modulation among MNs

during the four different RGM movements were relatively consistent, co-modulation among MNs during observation was less consistent, and co-modulation of AE neurons during execution even less so.

## Discussion

As neurophysiological studies have advanced from examination of single neurons to neuron populations, analytic approaches have advanced from analyses of single neuron firing rates to analyses of co-modulation patterns among neuron populations. The co-modulation in a neuronal population can be expressed as the trajectory of the simultaneous firing rates of the *N* neurons through their *N*-dimensional state space, and the predominant patterns of co-modulation can be extracted by projecting this high-dimensional trajectory into a low-dimensional subspace that captures a large proportion of the population's firing rate variance. Previous studies of reaching movements have shown that the low-dimensional population trajectories of PMd and M1 neurons occupy one subspace during a preparatory delay epoch and then transition to a different subspace during the reaching movement per se (*Kaufman et al., 2014*; *Elsayed et al., 2016*). Compared to reaching movements, however, the low-dimensional trajectories of neuronal activity controlling hand movements are relatively complex (*Rouse and Schieber, 2018*; *Suresh et al., 2020*). To approach this problem, rather than examining neural trajectories in subspaces that capture only a selected epoch of the behavioral task, we identified time series of instantaneous, condition-dependent subspaces covering the entire time course of RGM behavioral trials that included a preparatory delay epoch.

Using this approach, we found that the instantaneous, condition-dependent subspace of PM MN populations shifts progressively during both execution and observation of RGM trials. The instantaneous subspace of AE neuron populations likewise shifts progressively during action execution. This progressive shifting of the instantaneous subspace resembles that found previously using fractional overlap of condition-dependent variance in M1 neuron populations performing a similar RGM task without a delay epoch (*Rouse and Schieber, 2018*). Although the progressive shifting described here is a rotation in the mathematical sense, it is not necessarily a smooth rotation in a few dimensions. We therefore have used the word 'shift' to contrast with the smooth rotation of neural trajectories in a low-dimensional subspace described in other studies, particularly those using jPCA (*Churchland et al., 2012*; *Russo et al., 2020*; *Rouse et al., 2022*).

### Features of the instantaneous subspace

Short bursts of 'signal' related discharge are known to occur in a substantial fraction of PMd neurons beginning at latencies of ~60 ms following an instructional stimulus (*Weinrich et al., 1984*; *Cisek and Kalaska, 2004*). Here, we found that the instantaneous subspace shifted briefly toward the subspace present at the time of instruction onset (I), similarly during execution and observation. This brief trough in principal angle (*Figure 4A*) and the corresponding peak in classification accuracy (*Figure 7A*) in part may reflect smoothing of firing rates with a 50 ms Gaussian kernel. We speculate, however, that the early rise of this peak at the time of instruction onset also reflects the anticipatory activity often seen in PMd neurons in expectation of an instruction, which may not be entirely nonspecific, but rather may position the neural population to receive one of a limited set of potential instructions (*Mauritz and Wise, 1986*). We attribute the relatively low amplitude of peak classification accuracy for instruction trajectory segments to the likely possibility that only the last 40 ms of our 100 ms instruction segments captured signal-related discharge.

The firing rates of MNs in both PMv and PMd have been shown previously to modulate during preparatory delay periods (*Cisek and Kalaska, 2004*; *Maranesi et al., 2014*). During execution of a reaching task, condition-dependent subspaces during the preparatory delay are orthogonal to those found during the subsequent movement epochs (*Kaufman et al., 2014*; *Elsayed et al., 2016*). Studies that have identified such orthogonal subspaces specifically optimized preparatory and movement subspaces to be orthogonal to one another, however, whereas the present approach did not. Here, we found that during the preparatory delay epoch of the present RGM task, the condition-dependent, instantaneous subspace did not remain orthogonal to that which would be present at movement onset or during the movement epoch. Rather, as the preparatory delay proceeded, the instantaneous subspace shifted concurrently toward both the subspace that would be present at the time of the

go cue ending the preparatory delay (G) and that which would be present at movement onset (M). By time G, the instantaneous subspace had already shifted approximately halfway toward the time M subspace. This difference in the orthogonality of preparatory versus movement subspaces may reflect differences in reaching without grasping, which involves coordinated motion in 4 DOFs at the shoulder and elbow, versus the present RGM movements, which involve simultaneous, fluidly coordinated motion in at least 22 DOFs of the shoulder, elbow, wrist, and digits (*Rouse and Schieber, 2015*). Finally, we note that the progressive shift toward the subspace present at the onset of the final hold (H) did begin only after the delay period had ended (G) and around the time of movement onset (M).

## PM MN populations during execution versus observation

In general, instantaneous execution subspaces were distinct from instantaneous observation subspaces, indicated by the continuously large principal angles between them (*Figure 7*) and by low classification accuracy when execution trajectories were cross-projected into observation subspaces and vice versa (*Figure 7—figure supplement 1*). This was the case not only during corresponding time points in execution and observation trials, but throughout their entire time course. Moreover, in all three monkeys, progressive shifting of the instantaneous, condition-dependent subspace was absent both in the principal angles between execution and observation subspaces and in the decoding of execution trajectory segments cross-projected into observation subspaces (and vice versa). These findings indicate that the predominant modes of co-modulation among PM MNs are largely distinct during execution and observation.

Although MNs originally were thought to provide highly congruent neural representations of action execution and action observation (*Gallese et al., 1996*; *Rizzolatti et al., 1996*), the present findings are consistent with recent studies that have emphasized the considerable fraction of neurons with noncongruent activity, as well as differences in neural population activity during action execution versus action observation (*Jiang et al., 2020*; *Pomper et al., 2023*). As more situations have been investigated, the number of conditions needed to define a 'true' MN in the strict sense of being entirely congruent has grown, making the duration of such congruence brief and/or its likelihood comparable to chance (*Papadourakis and Raos, 2019*; *Pomper et al., 2023*).

We did not attempt to classify neurons in our PM MN populations as strictly congruent, broadly congruent, or noncongruent. Nevertheless, the minimal overlap we found in instantaneous execution and observation subspaces would be consistent with a low degree of congruence in our PM MN populations. Particularly during one session, monkey T was an exception in this regard, showing a considerable degree of overlap between execution and observation subspaces, not unlike the shared subspace found in other studies that identified orthogonal execution and observation subspaces as well (*Jiang et al., 2020*). Although our microelectrode arrays were placed in similar cortical locations in the three monkeys, by chance, monkey T's PM MN population may have included a substantial proportion of congruent neurons.

## Alignment of trajectory segments with canonical correlation

Given the complexity of condition-dependent neural trajectories across the entire time course of RGM trials (*Figure 3B*), rather than attempting to align entire neural trajectories, we applied canonical correlation to trajectory segments clipped for 100 ms following four well-defined behavioral events: instruction onset, go cue, movement onset, and the beginning of the final hold. In all cases, alignment was poorest for instruction segments, somewhat higher for go segments, and strongest for movement and hold segments (*Figure 8C*). This progressive increase in alignment likely reflects a progressive increase in the difference between average neuron firing rates for trials involving different objects (*Figure 5*) relative to the trial-by-trial variance in firing rate for a given object.

Corresponding neural representations of action execution and observation during task epochs with higher neural firing rates have been described previously in PMd MNs and in PMv MNs using representational similarity analysis (*Papadourakis and Raos, 2019*). And during force production in eight different directions, neural trajectories of PMd neurons draw similar 'clocks' during execution, cooperative execution, and passive observation (*Pezzulo et al., 2022*). Likewise, in the present study, despite execution and observation trajectories progressing through largely distinct subspaces, in all three monkeys, execution and observation trajectory segments showed some degree of alignment,

particularly the movement and hold segments (*Figure 8C*), indicating similar relationships among the latent dynamic representations of the four RGM movements during execution and observation.

Alignment between trajectory segments of the same PM MN population during execution versus observation in the same session, however, was less than that found between MN execution segments from two different sessions in the same monkey. In part, this may reflect the lower firing rates of PM MNs typically found during observation as compared to execution trials (*Ferroni et al., 2021*). Alternatively, the lower alignment may reflect more trial-by-trial variability in MN observation segments than in MN execution segments, as indicated by the limited within-group alignment of MN observation trajectory segments (*Figure 8D*).

Based on the assumption that AE neurons and MNs function as a homogenous neuron population during action execution, we had expected AE and MN execution trajectory segments to align closely. During execution trials, the progression of instantaneous condition-dependent subspaces and of classification accuracy in AE populations was quite similar to that in MN populations. We were surprised to find, therefore, that alignment between execution trajectory segments from AE populations and from the simultaneously recorded MN populations was even lower than alignment between MN execution and observation segments (*Figure 8C*, blue versus red). Moreover, whereas within-group alignment of MN execution trajectory segments was high, within-group alignment of AE neuron execution trajectory segments was low (*Figure 8D*, gray versus light blue). These findings indicate that the predominant patterns of co-modulation among MNs during execution are quite consistent within sessions, but the patterns of co-modulation among AE neurons are considerably more variable. Together with our previous finding that modulation of MNs leads that of non-MNs in time—both at the single neuron level and at the population level (*Mazurek and Schieber, 2019*)—this difference in consistency versus variability leads us to speculate that, during action execution, MNs carry a consistent forward model of the intended movement, while AE neurons carry more variable feedback information.

## The role of MN populations

Neither the congruence versus noncongruence of individual MN discharge nor the canonical correlation of population dynamics during execution and observation provides direct causal evidence that MNs mediate understanding of the observed actions of other individuals (*Hickok, 2009*; *Yuste, 2015*; *Krakauer et al., 2017*). Many interpretations of such findings are possible, and testing various hypotheses ultimately may require selective experimental manipulation (e.g. inactivation) of MN activity during observation in ways beyond our current capabilities. Nevertheless, the common finding that large fractions of neurons in both PM and M1 discharge both during execution and during observation makes it unlikely that the discharge of MNs during observation is vestigial, with no meaning for the organism.

Although we did not track extraocular movements, video monitoring demonstrated that our monkeys remained attentive throughout the blocks of observation trials, actively scanning the visual environment. Though perhaps not following the experimenter's movements closely with eye movements, or even with covert visual attention, the present results in and of themselves demonstrate that during observation trials the PM MN population was processing information on the sequential epochs of the behavioral task (*Mazurek et al., 2018*), as well as the object to which the experimenter's actions were directed on each trial. These findings are consistent with the notion that the PM MN population predictively represents the sequence of behavioral events during observation trials (*Kilner et al., 2007*; *Maranesi et al., 2014*; *Ferroni et al., 2021*). Our finding that within-group alignment of MN observation trajectory segments was lower than that of MN execution segments (*Figure 8D*), however, indicates more trial-by-trial variability of MN co-modulation during observation than during execution. In addition to any consistent, predictive, forward model of the observed experimenter's expected performance, MNs thus may also receive visual input that incorporates more variable, trial-by-trial deviation from the predicted performance being observed.

One classic interpretation of similar latent dynamics in the PM MN population during execution and observation would be that this similarity provides a means for the brain to recognize similar movements performed by the monkey during execution and by the experimenter during observation. Through some process akin to a communication subspace (*Semedo et al., 2019*), brain regions beyond PM might recognize the correspondence between the latent dynamics of the executed and observed actions.

Alternatively, given that observation of another individual can be considered a form of social interaction, PM MN population activity during action observation, rather than representing movements made by another individual similar to one's own movements, instead may represent different movements one might execute oneself in response to those made by another individual (*Ninomiya et al., 2020*; *Bonini et al., 2022*; *Ferrucci et al., 2022*; *Pomper et al., 2023*). This possibility is consistent with the finding that the neural dynamics of PM MN populations are more similar during observation of biological versus non-biological movements than during execution versus observation (*Albertini et al., 2021*). Though neurons active only during observation of others (AO units) have been hypothesized to drive observation activity in MNs, the present AO populations were too small to analyze with the approaches we applied here. Nevertheless, the similar relative organization of the execution and observation population activity in PM MNs revealed here by alignment of their latent dynamics through CCA could constitute a correspondence between particular movements that might be made by the subject in response to particular movements made by the other individual, i.e., responsive movements which would not necessarily be motorically similar to the observed movements.

The present analyses, as well as others, have focused on the condition-dependent variance in MN population activity (*Jiang et al., 2020*). Other studies that have not separated the condition-dependent versus condition-independent variance in neural activity have described even more similar latent dynamics during execution and observation (*Mazurek et al., 2018*; *Jerjian et al., 2020*; *Pezzulo et al., 2022*). We speculate that condition-dependent activity may represent particular types of movement (e.g. sphere, button, coaxial cylinder, or perpendicular cylinder) in a manner that differs depending on the actor (one's self versus another individual). Concurrently, condition-independent activity may provide a neural representation of a class of action (e.g. RGM movements) independent of the actor.

## Methods

### Key resources table

| Reagent type (species) or resource | Designation | Source or reference | Identifiers | Additional information |
|---|---|---|---|---|
| Software, algorithm | Trellis | Ripple | https://rippleneuro.com/support/software-downloads-updates/ | |
| Software, algorithm | MATLAB | MathWorks | https://www.mathworks.com/products/matlab.html | |
| Software, algorithm | | Custom code for data analysis | https://github.com/ShiftingSubspace/shiftsubs | |
| Other | Floating Microelectrode Arrays | Microprobes for Life Sciences | https://www.microprobes.com/products/multichannel-arrays/fma | See Neuron recording |
| Other | Trek | Ripple | https://rippleneuro.com/ripple-products/trek-electrophysiology-system/ | See Neuron recording |

Three Rhesus monkeys, R, T, and F (a 6 kg female, a 10 kg male, and an 11 kg male, *Macaca mulatta*) were used in the present study. All procedures for the care and use of these nonhuman primates followed the Guide for the Care and Use of Laboratory Animals and were approved by the University Committee on Animal Resources at the University of Rochester, Rochester, New York, under protocol #101058.

### Execution trials

Each monkey was trained to perform a delayed-response RGM task (*Figure 2*). Prior to each trial, a ring of blue LEDs was illuminated around the pole supporting a center object, and a 4 kHz tone began, both signaling the end of an inter-trial interval and the opportunity to begin a new trial. The monkey initiated the following sequence by pulling the center object for an initial hold epoch of randomly varied duration (500–1000 ms). A ring of blue LEDs around the pole supporting one of four peripheral objects was then illuminated, instructing the monkey as to the target object for the current trial. After 500 ms, these instruction LEDs were extinguished, and the monkey was required to wait for a preparatory delay epoch lasting randomly 500–2000 ms. At the end of this preparatory delay epoch, the blue LEDs for the center object were extinguished and the 4 kHz tone ceased, providing a go cue.

The monkey then reached for, grasped, and manipulated the remembered target object—turning a sphere, pushing a button, pulling a coaxial cylinder (coax), or pulling a perpendicular cylinder (perp). The RGM sequence was performed as a single, uninterrupted, fluid movement of the entire upper extremity (*Rouse and Schieber, 2015*; *Rouse and Schieber, 2016a*; *Rouse and Schieber, 2016b*). Once the instructed object had been manipulated, a ring of green LEDs around the object illuminated (indicating successful manipulation of the object) and the ring of blue LEDs for that object also illuminated (indicating correct object). The monkey was then required to hold the instructed object in its manipulated position for a final hold epoch of 1000 ms, after which the blue LEDs were extinguished. (The green LEDs extinguished whenever the monkey released the object.) After a 300 ms delay, the monkey received a liquid reward on each successful trial.

The selection and sequence of target objects in successive trials were controlled by custom software (Unified Task Control System, Gil Rivlis, https://github.com/grivlis/UTCS3_RGM), which also (1) generated behavioral event marker codes (*Figure 2B*) and (2) arranged trials involving the four different objects in a pseudorandom block design (*Rivlis, 2025*). The behavioral event marker codes indicated the times at which specific behavioral events occurred: start of trial, instruction onset, instruction offset, go cue (delay epoch ended), movement onset, hold began, hold ended, end of trial. One trial involving each of the four different objects was presented sequentially in a block. Once a block had been completed, the sequence of the four objects was shuffled randomly for the next block. To prevent the monkey from skipping more difficult objects, if the monkey failed to complete a trial successfully, the same target was repeated until the monkey succeeded.

## Observation trials

In a separate block of trials, the monkey observed an experimenter performing the same delayed-response RGM task. The experimenter occasionally made errors intentionally. The monkey received a reward each time the experimenter performed a successful trial, but not when the experimenter made an error, which kept the monkey attentive to the experimenter's performance. Although extraocular movements were not recorded or controlled, video monitoring verified that the monkey remained alert and attentive throughout blocks of observation trials.

## Neuron recording

Each of the three monkeys was implanted with FMAs (Microprobes for Life Sciences) in the PMv and the PMd. In monkeys R and T, 16-channel FMAs were implanted; in monkey F, 32-channel FMAs were used (*Figure 2C*). Monkeys R and F each had a total of 64 recording electrodes implanted in PMd and 64 in PMv, whereas monkey T had 64 in PMd, but only 48 in PMv. Broadband signals were recorded simultaneously from all 128 electrodes using a Trek/Trellis data acquisition system (Ripple, Salt Lake City, UT, USA), which also recorded the behavioral event marker codes generated by the behavioral control system. In each recording session, data were collected during similar numbers of successful trials involving each target object during execution and then during observation, as summarized in *Table 2*. Off-line, spike waveforms were extracted and sorted using custom software (*Rouse and Schieber, 2016a*). Sorted units were classified as definite single units, probably single units, multiunits, or noise based on their signal-to-noise ratio and estimated fraction of false-positive spikes using our previously published criteria. All three types of units were included in the present analyses.

## MN identification

Although many studies have focused on neurons from either PMv or PMd, given that neurons in each area have been shown to be modulated during both reaching and grasping (*Stark et al., 2007*) and during both execution and observation (*Papadourakis and Raos, 2019*), we chose to combine units from these two cortical areas for the present analyses. Each unit was tested for task-related modulation. Because a given neuron's firing rates during execution and observation trials almost always differed (*Ferroni et al., 2021*; *Pomper et al., 2023*), we tested each unit for modulation using data from these two contexts separately. Spike counts from each successful behavioral trial were extracted during eleven 200 ms periods: (i) before instruction onset, (ii) after instruction onset, (iii) before instruction offset, (iv) after instruction offset (delay epoch began), (v) before delay ended, (vi) after delay ended (reaction epoch began), (vii) before movement onset, (viii) after movement onset (movement epoch began), (ix) before movement ended, (x) after movement ended (hold epoch began), (xi)

before hold ended. We then conducted two-way ANOVA on these spike counts using object and time period as factors. We considered a unit task-related if it showed a significant main effect of either (i) object or (ii) time period, or a significant (iii) interaction effect. Any unit modulated significantly both during execution and during observation was considered to be a MN. Because each unit thus had six opportunities to show significance, we used a corrected significance criterion of $p<0.0083$ ($<0.05/6$). Any unit modulated during execution but not during observation was considered an action execution (AE) neuron. Any unit modulated during action observation but not during execution was considered an action observation neuron (AO). Units unmodulated during both execution and observation were considered not significantly (NS) related to the task.

## Data analysis

All data analysis was performed in MATLAB using custom code (https://github.com/ShiftingSubspace/shiftsubs; *Zhao, 2024*). Spike times for each neuron were binned (bin width = 1 ms), smoothed with a Gaussian kernel ($\sigma=50$ ms), and square-root transformed to render variance similar from low to high firing rates (*Kihlberg et al., 1972*; *Snedecor and Cochran, 1980*). The activity of each neuron was time-aligned to four behavioral events and truncated before and after using the median delay, reaction, and movement times per object and per session as follows: (i) instruction onset (I)—500 ms before to 500 ms after; (ii) go cue (G)—median delay duration before to half the median reaction time after; (iii) movement onset (M)—half the median reaction time before to 200 ms after; and (iv) start of final hold (H)—200 ms before to 200 ms after. These four snippets of neural activity were concatenated for each trial. Neural activity was then stored as a three-dimensional tensor ($N \times K \times T$, where $N$ is the number of neurons, $K$ the number of trials, and $T$ the number of time points) for each of the four target objects.

## Instantaneous subspace identification

Instantaneous neural subspaces were identified at 1 ms intervals. At each 1 ms time step, the $N$-dimensional neural firing rates from trials involving the four different objects—sphere, button, coaxial cylinder, and perpendicular cylinder—were averaged separately, providing four points in the $N$-dimensional space representing the average neural activity for trials involving the different objects at that time step. PCA then was performed on these four points. Because three dimensions capture all the variance of four points, three principal component dimensions fully defined each instantaneous subspace. Each instantaneous 3D subspace can be considered a filter described by a matrix, $W$, that can project high-dimensional neural activity into a low-dimensional subspace, with the time series of instantaneous subspaces, $W_i$, forming a time series of filters (*Figure 1B*).

## Trajectory visualization and separation

We projected 100 ms segments of neural activity into each instantaneous subspace by multiplying the neural activity, $X(t)$, by the transforming matrix for the $i$th subspace, $W_i$, which yielded low-dimensional trajectory segments, $L(t) = X(t) W_i$ ($t \in T$). This process was repeated for each instantaneous subspace in the time domain of interest. To quantify the separation between the four trial-averaged trajectory segments involving the different objects in a given instantaneous subspace, we then calculated their cumulative separation ($CS$) as:

$$CS = \frac{1}{T} \sum_{t \in T} D(t) = \frac{1}{T} \sum_{t \in T} \sum_{i \neq j} d_{ij}(t)$$

where $d_{ij}(t)$ is the three-dimensional Euclidean distance between the $i$th and $j$th trajectories at time point $t$. We summed the six pairwise distances between the four trajectory segments across time points and normalized by the number of time points, $T = 100$. The larger the $CS$, the greater the separation of the trajectory segments.

## Subspace comparisons—principal angles

To assess the progressive shift of instantaneous subspaces, we computed the principal angles (*Björck and Golub, 1973*; *Gallego et al., 2018*) between the instantaneous subspace at each of four selected time points—onset of the instruction (I), go cue (G), onset of movement (M), and beginning of the

final hold (H)—and each of the other instantaneous subspaces in a time series. For example, given the three-dimensional instantaneous subspace at the time of movement onset, $W_M$, and at any other time, $W_i$, we calculated their 3×3 inner product matrix and performed singular value decomposition to obtain:

$$W_M^T W_i = P_M C P_i^T$$

where 3×3 matrices $P_M$ and $P_i$ define new manifold directions which successively minimize the three principal angles specific to the two subspaces being compared. The elements of the diagonal matrix $C$ are then the ranked cosines of the principal angles, $\theta_i$, ordered from smallest to largest:

$$C = diag\left(\cos\left(\theta_1\right), \cos\left(\theta_2\right), \cos\left(\theta_3\right)\right)$$

In *Figure 4—figure supplement 1*, using all trials from monkey R, session 1, we have plotted the three principal angles as a function of time. Note that at the time when $W_i = W_M$, all three principal angles are zero by definition, and the sharp decline before time M and the sharp rise afterward reflect the Gaussian kernel ($\sigma$=50 ms) used to smooth unit firing rates. These sharp troughs thus are trivial, but both the gradual decline before and the gradual rise following the sharp troughs are not. Given that the set of three principal angles typically followed similar time courses, in the Results, we illustrate only the first principal angle, $\theta_1$.

Furthermore, to provide some indication of the degree of variability in the first principal angle, we randomly selected 20 trials involving each target object (totaling 80 trials) with replacement and calculated the first principal angle as a function of time, repeating this process 10 times. The results, shown in *Figures 4 and 7*, *Figure 4—figure supplement 3*, and *Figure 7—figure supplement 2*, are presented as the mean ± 1 standard deviation across these 10-fold cross-validations. Note that this mean never reaches zero because the instantaneous subspaces at times I, G, M, and H were computed using all the available trials.

In the example of *Figure 4—figure supplement 1*, the first principal angle never reached 90° either. To determine whether this reflected a lack of orthogonality or a limitation of population size, we computed the first principal angle between a fixed three-dimensional subspace and 5000 three-dimensional subspaces randomly chosen from $N$-dimensional spaces, for $N$ varying from 5 to 500. *Figure 4—figure supplement 2* shows that for large $N$, principal angles between a fixed subspace and other randomly chosen subspaces are likely to be close to 90°. But as $N$ decreases, these random principal angles are less likely to approach 90°, without necessarily indicating nonrandom overlap of the subspaces. In *Figures 4 and 7*, *Figure 4—figure supplement 3*, and *Figure 7—figure supplement 2*, we therefore indicate levels of principal angles that might arise by chance alone using the smallest $N$ from any of the three sessions for a given monkey (see *Table 2*).

## Decodable information—LSTM

As illustrated schematically in *Figure 1B*, the same segment of high-dimensional neural activity projected into different instantaneous subspaces can generate low-dimensional trajectories of varying separation. The degree of separation among the projected trajectory segments will depend not only on their separation at the time when the segments were clipped, but also on the similarity of the subspaces into which the trajectory segments are projected. To quantify the combined effects of trajectory separation and projection into different subspaces, we projected high-dimensional neural trajectory segments (each including 100 points at 1 ms intervals) from successful trials involving each of the four different target objects into time series of three-dimensional instantaneous subspaces at 50 ms intervals. In each of these instantaneous subspaces, the neural trajectory segment from each trial thus became a 100 time point × three-dimensional matrix. For each instantaneous subspace in the time series, we then trained a separate LSTM (*Hochreiter and Schmidhuber, 1997*) classifier to attribute each of the neural trajectories from individual trials to one of the four target object labels: sphere, button, coaxial cylinder, or perpendicular cylinder. Using MATLAB's Deep Learning Toolbox, each LSTM classifier had 3 inputs (instantaneous subspace dimensions), 20 hidden units in the bidirectional LSTM layer, and a softmax layer preceding the classification layer which had 4 output classes (target objects). The total number of successful trials available in each session for each object is given in *Table 1*. To avoid bias based on the total number of successful trials, we used the minimum number of successful trials across the four objects in each session, selecting that number

from the total available randomly with replacement. Each LSTM classifier was trained with MATLAB's adaptive moment estimation (Adam) optimizer on 40% of the selected trials, and the remaining 60% were decoded by the trained classifier. The success of this decoding was used as an estimate of classification accuracy from 0 (no correct classifications) to 1 (100% correct classifications). This process was repeated 10 times, and the mean ± standard deviation across the 10 folds was reported as the classification accuracy at that time. Classification accuracy of trials projected into each instantaneous subspace at 50 ms intervals was plotted as a function of trial time.

### Similarity of aligned latent dynamics

We used CCA to compare the similarity of latent dynamics in different subspaces (*Gallego et al., 2020*). In brief, given latent dynamics (trajectory segments) in two original subspaces, $L_A$ and $L_B$, CCA finds a linear transformation of each original subspace such that, when projected into a common subspace, the aligned latent dynamics, $\tilde{L}_A$ and $\tilde{L}_B$, are maximally correlated in each dimension of the common subspace. Larger canonical correlation coefficient (CCs) indicate a higher degree of alignment.

CCA was performed as follows: The original latent dynamics, $L_A$ and $L_B$, were first transformed and decomposed as $L_A^T = Q_A R_A$ and $L_B^T = Q_B R_B$. The first $m$=3 column vectors of each $Q_i$ provide an orthonormal basis for the column vectors of $L_i^T$ (where $i = A, B$). Singular value decomposition on the inner product matrix of $Q_A$ and $Q_B$ then gives $Q_A^T Q_B = USV^T$, and new manifold directions that maximize pairwise correlations are provided by $M_A = R_A^{-1} U$ and $M_B = R_B^{-1} V$. We then projected the original latent dynamics into the new, common subspace: $\tilde{L}_A^T = L_A^T M_A$; $\tilde{L}_B^T = L_B^T M_B$. Pairwise CCs between the aligned latent dynamics sorted from largest to smallest are then given by the elements of the diagonal matrix $S = \tilde{L}_A \tilde{L}_B^T$.

To provide an estimate of variability, we used a bootstrapping approach to CCA. From each of two datasets, we randomly selected 20 trials involving each target object (totaling 80 trials) with replacement, clipped trajectory segments from each of those trials for 100 ms (100 points at 1 ms intervals) after the instruction onset, go cue, movement onset, or beginning of the final hold, and performed CCA as described above. (Note that because session 1 from monkey R included only eight button trials [*Table 1*], we excluded this session from CCA analyses.) With 500 iterations, we obtained a distribution of the CCs between the two datasets in each of the three dimensions of the aligned subspace, which permitted statistical comparisons. We then used this approach to evaluate alignment of latent dynamics between different sessions (e.g. execution trials on two different days), between different contexts (e.g. execution and observation), and between different neural populations (e.g. MNs and AE neurons). This bootstrapping approach further enabled us to assess the consistency of relationships among neural trajectories within a given group—i.e. the same neural population during the same context (execution or observation) in the same session—by drawing two separate random samples of 80 trials from the same population, context, and session (*Figure 8D*), which would not have been possible had we concatenated trajectory segments from all trials in the session (*Gallego et al., 2020*; *Safaie et al., 2023*).

## Acknowledgements

The authors thank John Housel and Jennifer Gardinier for technical assistance, Gil Rivlis for custom task-control software, and Marsha Hayles for editorial comments. This work was supported by grant R01NS102343 (MHS) from the National Institute of Neurological Disorders and Stroke.

## Additional information

### Funding

| Funder | Grant reference number | Author |
|---|---|---|
| National Institute of Neurological Disorders and Stroke | R01NS102343 | Marc H Schieber |

| Funder | Grant reference number | Author |
|---|---|---|

The funders had no role in study design, data collection and interpretation, or the decision to submit the work for publication.

## Author contributions

Zhonghao Zhao, Software, Formal analysis, Validation, Visualization, Methodology, Writing - original draft; Marc H Schieber, Conceptualization, Resources, Data curation, Supervision, Funding acquisition, Investigation, Visualization, Methodology, Project administration, Writing – review and editing

## Author ORCIDs

Marc H Schieber ⬡ https://orcid.org/0000-0002-4315-8161

## Ethics

All procedures for the care and use of these non-human primates followed the Guide for the Care and Use of Laboratory Animals and were approved by the University Committee on Animal Resources at the University of Rochester, Rochester, New York under protocol #101058.

Reviewer #2 (Public review): https://doi.org/10.7554/eLife.94165.3.sa1
Reviewer #3 (Public review): https://doi.org/10.7554/eLife.94165.3.sa2
Reviewer #4 (Public review): https://doi.org/10.7554/eLife.94165.3.sa3
Author response https://doi.org/10.7554/eLife.94165.3.sa4

# Additional files

## Supplementary files

MDAR checklist

## Data availability

Processed data for each of the 9 recording sessions analyzed in this study are publicly available at https://doi.org/10.5061/dryad.cvdncjtfq. The raw data for all 9 sessions requires ~1.5 Tb of storage space and therefore have not been shared in their entirety on a publicly accessible server. However, as an example, the raw data files from one of the 9 recording sessions (monkey R, session 2) also are publicly available at https://doi.org/10.5061/dryad.cvdncjtfq. Upon request, the remainder of the raw data files are available from the lead contact, Marc H. Schieber (mschiebe@ur.rochester.edu). Files will be sent on an external hard drive. No application or project proposal is required. Also available at https://doi.org/10.5061/dryad.cvdncjtfq are the processed data used for Figure 8, panels C and D, and for Figure 5, figure supplement 1. Other figures were generated from the processed data files using the analysis code packages available as described below. Code packages for all analyses performed in this work are available at: https://github.com/ShiftingSubspace/shiftsubs (copy archived at *Schieber, 2024*). Code for the custom task control software used in this study is available at: https://github.com/grivlis/UTCS3_RGM (copy archived at *grivlis, 2025*).

The following dataset was generated:

| Author(s) | Year | Dataset title | Dataset URL | Database and Identifier |
|---|---|---|---|---|
| Zhao Z, Schieber M | 2025 | Data from: Progressively shifting patterns of co-modulation among premotor cortex neurons carry dynamically similar signals during action execution and observation | https://doi.org/10.5061/dryad.cvdncjtfq | Dryad Digital Repository, 10.5061/dryad.cvdncjtfq |

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
