## [Editor Report · eLife Assessment]

This **valuable** study reports on the characteristics of premotor cortical population activity during the execution and observation of a moderately complex reaching and grasping task. By using new variants of well-established techniques to analyse neural population activity, the authors provide **solid** evidence that while the geometry of neural population activity changes between execution and observation, their dynamics are largely preserved. Although these findings are novel and robust, pending additional controls and analyses, the authors should further clarify the functional implications of their findings.

---

## [Referee Report · Reviewer #2 (Public review)]

The authors investigated the similarity (or lack thereof) of neural dynamics while monkeys reached to and manipulated one of 4 objects in each trial, compared to observing similar movements performed by experimenters. They focused on mirror neurons (MNs) and rather convincingly showed that MNs dynamics are dissimilar during executing vs. observing actions. The manuscript has improved quite significantly compared to the previous version and I congratulate the authors for that. However, there are still a few points I would like to raise that I think will improve the manuscript scientifically and make it more pleasant to read.

- I appreciate the nicely compiled literature review which provides the context for the manuscript.

- Message: The takeaway message of the paper is inconsistent and changes throughout the paper. To me, the main takeaway is that observation and execution subspaces progress during the trial (Fig 4), and that they are distinct processes and rather dissimilar, as stated in #440-441, #634-635, etc. But the title of the paper implies the opposite. Some of the interpretations of the results (e.g., Fig 8) also imply similarity of dynamics.

- Readability: I have many issues with the readability/organisation of the paper. Unfortunately, I still find the quality of data presentation low. Below I list a few points:

(1) In 5 sessions out of 9, there are fewer than 20 neurons categorised as AE. This means this population is under-sampled in the data which makes applying any neural population techniques questionable. Moreover, the relevance of the AE analysis is also sometimes unclear: In Fig 4, the AE-related panels are just referred to once in the paper. Yet AE results are presented right next to the main results throughout the paper.

(2) Figures are low resolution and pixelated. There are some faded horizontal and vertical lines in Fig1B that are barely visible. Moreover, it may be my personal preference, but I think Fig1 is more confusing than helpful. Although panel A shows some planes rotating, indicating time-varying dynamics, I couldn't understand what more panel B is trying to convey. The arrow of time is counterclockwise, but the planes progress clockwise (i > ii > iii). Similarly, panel C just seems to show some points being projected to orthogonal subspaces (even though later in the paper we'll see that observation and execution subspaces are not orthogonal), and the CCA subspace illustrated in the same high-d space, which mathematically may be inaccurate, as CCA projects the data to a new space.

In Fig 2A, the objects are too small and pixelated as well. I suggest an overhaul of the figures to make the paper more accessible.

(3) Clarity of the text: The manuscript text could be more concise, to the point, avoiding repetitions, self-consistent, and simply readable. To name a few issues: Single letter acronyms were used to refer to trial epochs (I/G/M/H). M alone has been re-defined 13 different times in the text as in: ...Movement (M)..., excluding every related figure. The acronym (I) refers to the instruction epoch, the high-d space in Fig 1, and panel I of some figures. The acronym MN for Mirror Neurons was defined 4 separate times in the text yet spelled out as Mirror Neuron more than 2 dozen times. CD is defined in the caption of Fig 3 and never used, despite condition-dependent being a common term in the text. Many sentences, e.g., "In contrast, throughout..." in #265-#269, and "To summarize,..." in #270-#275, are too long with difficult wording. To get the point from these sentences, I had to read them many times, and go back and forth between them and the figure. Rewriting such sentences makes the manuscript much more accessible.

- Figure 3: It appears that the condition independent signal has been calculated by subtracting the average of the 4 neural trajectories in Fig 3A, corresponding to different objects. Whereas #133 suggests that it should be calculated by subtracting the average firing rate of different conditions. Assuming I got the methods right, dynamics being "knotted" (#234) after removing the condition independent signal could be because they are similar, so subtracting the condition independent signal leaves us with the noise component. This matters for the manuscript especially since this is the reason for performing the more sensitive instantaneous subspaces.

- Decoding results: I appreciate that the authors improved the decoding results in this version of the manuscript. Now it is much more interesting. However oddly, it appears that only data from 1 monkey is shown. #370 says the results from the other 2 are similar. The decoding data from every monkey must be shown. If the results are similar, they must be at least in Supplements. Currently, only 1 session (out of 3) in the Observation condition seems to decode the object type. This effect, if consistent across animals and session, is very interesting on its own and challenges other claims in the paper.

- Figure8: I reiterate the issue #7 in my previous review. I appreciate the authors clearing some methods, but my concern persists. As per line #839, spiking activity has been smoothed with a 50ms kernel. Thus, unless trial data is concatenated, I suspect the 100ms window used for this analysis is too short (small sample size), thus the correlation values (CCs) might be spurious. References cited in this section use a smaller smoothing kernel (30ms) and a much longer window (~450ms).

Moreover, I don't know why the authors chose to show correlation values in 3D space! Values of Fig8C-red are impossible to know. Furthermore, the manuscript insists on CC values of the Hold period being high, which is probably correct. But I wonder why the focus on the Hold period? I think the most relevant epoch for analysing the MNs is the Movement where the actual action happens. Interestingly, in the movement epoch, the CC values are visibly low. The reason why Hold results are more important and why the CCs in Movement are so low should be clarified in the text. Especially, statements like that in #661 seem particularly unjustified.

---

## [Referee Report · Reviewer #3 (Public review)]

In their study, Zhao et al. investigated the population activity of mirror neurons (MNs) in the premotor cortex of monkeys either executing or observing a task consisting of reaching to, grasping, and manipulating various objects. The authors proposed an innovative method for analyzing the population activity of MNs during both execution and observation trials. This method enabled to isolate the condition dependent variance in neural data and to study its temporal evolution over the course of single trials. The method proposed by the authors consists of building a time series of "instantaneous" subspaces with single time step resolution, rather than a single subspace spanning the entire task duration. As these subspaces are computed on an instant time basis, projecting neural activity from a given task time into them results in latent trajectories that capture condition-dependent variance while minimizing the condition-independent one. Authors then analyzed the time evolution of these instantaneous subspaces and revealed that a progressive shift is present in subspaces of both execution and observation trials, with slower shifts during the grasping and manipulating phases compared to the initial preparation phase. Finally, they compared the instantaneous subspaces between execution and observation trials and observed that neural population activity did not traverse the same subspaces in these two conditions. However, they showed that these distinct neural representations can be aligned with Canonical Correlation Analysis, indicating dynamic similarities of neural data when executing and observing the task. The authors speculated that such similarities might facilitate the nervous system's ability to recognize actions performed by oneself or another individual.

Unlike other areas of the brain, the analysis of neural population dynamics of premotor cortex MNs is not well established. Furthermore, analyzing population activity recorded during non-trivial motor actions, distinct from the commonly used reaching tasks, serves as a valuable contribution to computational neuroscience. This study holds particular significance as it bridges both domains, shedding light on the temporal evolution of the shift in neural states when executing and observing actions. The results are moderately robust, and the proposed analytical method could potentially be used in other neuroscience contexts.

---

## [Referee Report · Reviewer #4 (Public review)]

Summary:

In this study, the authors explore the neural dynamics of mirror neurons in the premotor cortex, focusing on the relationship between neural activity during action execution and observation. The study presents a rich dataset from three monkeys, with recordings from two regions per monkey. The authors use a method to analyze instantaneous neural subspaces and track their temporal evolution. Consistent with prior literature, they report that execution and observation subspaces remain largely distinct throughout the trial. However, after applying canonical correlation analysis, they observe a notable alignment between execution and observation activities, suggesting the presence of shared neural codes. The study is well-designed, and the analyses are thoroughly documented, occasionally overly so in the main text. While most findings are compelling, I find the conclusions drawn from Figure 8 less convincing. Specifically, I am skeptical about the application of CCA in this context and the subsequent interpretations regarding execution-observation similarity, which is a central claim of the manuscript.

• The authors cite Safaie et al. 2023 as a precedent for applying CCA to align neural population dynamics. However, in that study, CCA was used to align neural dynamics across different animals, a justifiable approach given that neural trajectories exist in separate neural state spaces for each animal. Here, CCA is applied to align execution and observation activities within the same neural state space of the same MNs. I find this application of CCA less well-justified, as it may overestimate execution-observation similarity.

• The control conditions presented in Figures 8C and 8D are somewhat reassuring, as they show that the similarity introduced by CCA is not universally high. However, these controls appear to be limited to the Hold epoch. It remains unclear whether the same holds true for the Go and Movement epochs.

• In Figure 5, the authors display low-dimensional representations of four objects across task epochs during execution (A) and observation (B). The diagonals of the matrices reveal clear differences between execution and observation configurations across all four epochs. The authors suggest using CCA to align these configurations; however, this alignment seems to require time-specific application of CCA for each epoch (as demonstrated in Figure 8 for the Hold epoch). The need for time-specific adjustments likely depends on the fact that execution and observation subspaces are continuously shifting over time (as authors show in Figure 4), but this approach appears to be a strained attempt to demonstrate similarity between execution and observation codes.

• The authors themselves offer an alternative hypothesis (line 730): that "PM MN population activity during action observation, rather than representing movements made by another individual similar to one's own movements, instead may represent different movements one might execute oneself in response to those made by another individual". This interpretation appears more congruent with the data presented.

• In the end, I am left with a sense of ambiguity: which analysis should be considered more reliable, the negligible correspondence between execution and observation activity depicted in Figure 7, or the considerable similarity shown in Figure 8? The authors should address this apparent contradiction and provide a clearer discussion to reconcile these findings.

---

## [Author Response]

The following is the authors’ response to the original reviews.

Major changes in the revised manuscript include:

(1) The distinction between condition-dependent versus condition-independent variation in neural activity has been clarified.

(2) Principal angle calculations have been added.

(3) Neurons modulated during action execution but not during action observation have been analyzed to compare and contrast with mirror neurons.

(4) Canonical correlation analysis has been extended to three dimensions.

(5) Speculations have been moved to and modified in the Discussion.

(6) Computational details have been expanded in the Methods.

**Public Reviews:**

**Reviewer #1 (Public Review):**
Summary and strengths. This paper starts with an exceptionally fair and balanced introduction to a topic, the mirror neuron literature, which is often debated and prone to controversies even in the choice of the terminology. In my opinion, the authors made an excellent job in this regard, and I really appreciated it. Then, they propose a novel method to look at population dynamics to compare neural selectivity and alignment between execution and observation of actions performed with different types of grip.

Thank you.

Weakness.Unfortunately, the goal and findings within this well-described framework are less clear to me. The authors aimed to investigate, using a novel analytic approach, whether and to what extent a match exists between population codes and neural dynamics when a monkey performs an action or observes it performed by an experimenter. This motivation stems from the fact that the general evidence in the literature is that the match between visual and motor selectivity of mirror neuron responses is essentially at a chance level. While the approach devised by the author is generally well-described and understandable, the main result obtained confirms this general finding of a lack of matching between the two contexts in 2 out of the three monkeys. Nevertheless, the authors claim that the patterns associated with execution and observation can be re-aligned with canonical correlation, indicating that these distinct neural representations show dynamical similarity that may enable the nervous system to recognize particular actions. This final conclusion is hardly acceptable to me, and constitutes my major concern, at least without a more explicit explanation: how do we know that this additional operation can be performed by the brain?

Point taken. In the Discussion, we now have clarified that this is our speculation rather than a conclusion and we also offer an alternative interpretation (lines 724 to 744):

“One classic interpretation of similar latent dynamics in the PM MN population during execution and observation would be that this similarity provides a means for the brain to recognize similar movements performed by the monkey during execution and by the experimenter during observation. Through some process akin to a communication subspace (Semedo et al., 2019), brain regions beyond PM might recognize the correspondence between the latent dynamics of the executed and observed actions.

Alternatively, given that observation of another individual can be considered a form of social interaction, PM MN population activity during action observation, rather than representing movements made by another individual similar to one’s own movements, instead may represent different movements one might execute oneself in response to those made by another individual (Ninomiya et al., 2020; Bonini et al., 2022; Ferrucci et al., 2022; Pomper et al., 2023). This possibility is consistent with the finding that the neural dynamics of PM MN populations are more similar during observation of biological versus non-biological movements than during execution versus observation (Albertini et al., 2021). Though neurons active only during observation of others (AO units) have been hypothesized to drive observation activity in MNs, the present AO populations were too small to analyze with the approaches we applied here. Nevertheless, the similar relative organization of the execution and observation population activity in PM MNs revealed here by alignment of their latent dynamics through CCA could constitute a correspondence between particular movements that might be made by the subject in response to particular movements made by the other individual, i.e. responsive movements which would not necessarily be motorically similar to the observed movements.”

Is this a computational trick to artificially align something that is naturally non-aligned, or can it capture something real and useful?

We feel this is more than a trick. In the Introduction, we now have clarified (lines 166 to 170):

“Such alignment would indicate that the relationships among the trajectory segments in the execution subspace are similar to the relationships among the trajectory segments in the observation subspace, indicating a corresponding structure in the latent dynamic representations of execution and observation movements by the same PM MN population.”

In the Results we give the follow example (lines 446 to 455):

“Such alignment would indicate that neural representations of trials involving the four objects bore a similar relationship to one another in neural space during execution and observation, even though they occurred in different subspaces. For example, the trajectories of PMd+M1 neuron populations recorded from two different monkeys during center-out reaching movements could be aligned well (Safaie et al., 2023). CCA showed, for example, that in both brains the neural trajectory for the movement to the target at 0° was closer to the trajectory for movement to the target at 45° than to the trajectory for the movement to the target at 180°. Relationships among these latent dynamic representations of the eight movements thus were similar even though the neural populations were recorded from two different monkeys.”

And in the Discussion we now compare (lines 677 to 686):

“Corresponding neural representations of action execution and observation during task epochs with higher neural firing rates have been described previously in PMd MNs and in PMv MNs using representational similarity analysis RSA (Papadourakis and Raos, 2019). And during force production in eight different directions, neural trajectories of PMd neurons draw similar “clocks” during execution, cooperative execution, and passive observation (Pezzulo et al., 2022). Likewise in the present study, despite execution and observation trajectories progressing through largely distinct subspaces, in all three monkeys execution and observation trajectory segments showed some degree of alignment, particularly the Movement and Hold segments (Figure 8C), indicating similar relationships among the latent dynamic representations of the four RGM movements during execution and observation.”

Based on the accumulated evidence on space-constrained coding of others' actions by mirror neurons (e.g., Caggiano et al. 2009; Maranesi et al. 2017), recent evidence also cited by the authors (Pomper et al. 2023), and the most recent views supported even by the first author of the original discovery (i.e., Vittorio Gallese, see Bonini et al. 2022 on TICS), it seems that one of the main functions of these cells, especially in monkeys, might be to prepare actions and motor responses during social interaction rather than recognizing the actions of others - something that visual brain areas could easily do better than motor ones in most situations. In this perspective, and given the absence of causal evidence so far, the lack of visuo-motor congruence is a potentially relevant feature of the mechanism rather than something to be computationally cracked at all costs.

We agree that this perspective provides a valuable interpretation of our findings. In the Discussion, we have added the following paragraph (lines 730 to 744):

“Alternatively, given that observation of another individual can be considered a form of social interaction, PM MN population activity during action observation, rather than representing movements made by another individual similar to one’s own movements, instead may represent different movements one might execute oneself in response to those made by another individual (Ninomiya et al., 2020; Bonini et al., 2022; Ferrucci et al., 2022; Pomper et al., 2023). This possibility is consistent with the finding that the neural dynamics of PM MN populations are more similar during observation of biological versus non-biological movements than during execution versus observation (Albertini et al., 2021). Though neurons active only during observation of others (AO units) have been hypothesized to drive observation activity in MNs, the present AO populations were too small to analyze with the approaches we applied here. Nevertheless, the similar relative organization of the execution and observation population activity in PM MNs revealed here by alignment of their latent dynamics through CCA could constitute a correspondence between particular movements that might be made by the subject in response to particular movements made by the other individual, i.e. responsive movements which would not necessarily be motorically similar to the observed movements.”

Specific comments on Results/Methods:I can understand, based on the authors' hypothesis, that they employed an ANOVA to preliminarily test whether and which of the recorded neurons fit their definition of "mirror neurons". However, given the emphasis on the population level, and the consolidated finding of highly different execution and observation responses, I think it could be interesting to apply the same analysis on (at least also) the whole recorded neuronal population, without any preselection-based on a single neuron statistic. Such preselection of mirror neurons could influence the results of EXE-OBS comparisons since all the neurons activated only during EXE or OBS are excluded. Related to this point, the authors could report the total number of recorded neurons per monkey/session, so that also the fraction of neurons fitting their definition of mirror neuron is explicit.

We are aware that a number of recent studies from other laboratories already have analyzed the entire population of neurons during execution versus observation, without selectively analyzing neurons active during both execution and observation (Jiang et al., 2020; Albertini et al., 2021). However, our focus lies not in how the entire PM neural population encodes execution versus observation, but in the differential activity of the mirror neuron subpopulation in these two contexts. Our new Table 2 presents the numbers of mirror neurons (MN), action execution only neurons (AE), action observation only neurons (AO), and neurons not significantly task-related during either execution or observation (NS). Although we often recorded substantial numbers of AE neurons, very few AO neurons were found in our recordings. In analyzing the AE subpopulation, we found unexpected differences in canonical correlation alignment between and within the MN and AE neuron populations. In view of the editors’ comments that “…the reviewers provided several specific recommendations of new analyses to include. However, now the paper feels extremely long…”. We have chosen to focus on comparing AE neurons with MNs.

Furthermore, the comparison of the dynamics of the classification accuracy in figures 4 and 5, and therefore the underlying assumption of subspaces shift in execution and observation, respectively, reveal substantial similarities between monkeys despite the different contexts, which are clearly greater than the similarities among neural subspaces shifts across task epochs: to me, this suggests that the main result is driven by the selected neural populations in different monkeys/implants rather than by an essential property of the neuronal dynamics valid across animals. Could the author comment on this issue? This could easily explain the "strange" result reported in figure 6 for monkey T.

We have taken the general approach of emphasizing findings common across individual animals, but also reporting individual differences. We have added the following in the Discussion (lines 645 to 654):

“We did not attempt to classify neurons in our PM MN populations as strictly congruent, broadly congruent, or non-congruent. Nevertheless, the minimal overlap we found in instantaneous execution and observation subspaces would be consistent with a low degree of congruence in our PM MN populations. Particularly during one session monkey T was an exception in this regard, showing a considerable degree of overlap between execution and observation subspaces, not unlike the shared subspace found in other studies that identified orthogonal execution and observation subspaces as well (Jiang et al., 2020). Although our microelectrode arrays were placed in similar cortical locations in the three monkeys, by chance monkey T’s PM MN population may have included a substantial proportion of congruent neurons.”

**Reviewer #2 (Public Review):**
In this work, the authors set out to identify time-varying subspaces in the premotor cortical activity of monkeys as they executed/observed a reach-grasp-hold movement of 4 different objects. Then, they projected the neural activity to these subspaces and found evidence of shifting subspaces in the time course of a trial in both conditions, executing and observing. These shifting subspaces appear to be distinct in execution and observation trials. However, correlation analysis of neural dynamics reveals the similarity of dynamics in these distinct subspaces. Taken together, Zhao and Schieber speculate that the condition-dependent activity studied here provides a representation of movement that relies on the actor.This work addresses an interesting question. The authors developed a novel approach to identify instantaneous subspaces and decoded the object type from the projected neural dynamics within these subspaces. As interesting as these results might be, I have a few suggestions and questions to improve the manuscript:(1) Repeating the analyses in the paper, e.g., in Fig5, using non-MN units only or the entire population, and demonstrating that the results are specific to MNs would make the whole study much more compelling.

We have added analyses of those non-MNs modulated significantly during action execution but not during observation, which we refer to as AE neurons. The additional findings from these analyses are spread throughout the manuscript:

Lines 284-293:

“We also examined the temporal progression of the instantaneous subspace of AE neurons. As would be expected given that AE neurons were not modulated significantly during observation trials, in the observation context AE populations had no gradual changes in principal angle (Figure 4 – figure supplement 3). During execution, however, Figure 4I-L show that the AE populations had a pattern of gradual decrease in principal angle similar to that found in the MN population (Figure 4A-D). After the instruction onset, the instantaneous subspace shifted quickly away from that present at time I and progressed gradually toward that present at times G and M, only shifting toward that present at time H after movement onset. As for the PM MN populations, the condition-dependent subspace of the PM AE populations shifted progressively over the time course of execution RGM trials.”

Lines 411-419:

“During execution trials, classification accuracy for AE populations (Figure 6I-L) showed a time course quite similar to that for MN populations, though amplitudes were lower overall, most likely because of the smaller population sizes. During observation, AE populations showed only low-amplitude, short-lived peaks of classification accuracy around times I, G, M, and H (Figure 6 – figure supplement 1). Given that individual AE neurons showed no statistically significant modulation during observation trials, even these small peaks might not have been expected. Previous studies have indicated, however, that neurons not individually related to task events nevertheless may contribute to a population response (Shenoy et al., 2013; Cunningham and Yu, 2014; Gallego et al., 2017; Jiang et al., 2020).”

Lines 495-508:

“Although MNs are known to be present in considerable numbers in both the primary motor cortex and premotor cortex (see Introduction), most studies of movement-related cortical activity in these areas make no distinction between neurons with activity only during action execution (AE neurons) and those with activity during both execution and observation (MNs). This reflects an underlying assumption that during action execution, mirror neurons function in parallel with AE neurons, differing only during observation. We therefore tested the hypothesis that MN and AE neuron execution trajectory segments from the same session would align well. Figure 8C (blue) shows the mean CCs between MN and AE execution trajectory segments across 8 alignments (MN/AE; 2 R, 3 T, 3 F), which reached the highest values for the Hold segments \begin{document}$(\overline{C C 1}=0.57, \overline{C C 2}=0.35, \overline{C C 3}=0.19)$\end{document}. All three of these coefficients were substantially lower than those for the MN execution vs. observation alignments given above. Surprisingly, the alignment of AE neuron execution trajectory segments with those of the simultaneously recorded MN population was weaker than the alignment of MN trajectories during execution vs. observation.

Did these differences in MN:1/2, MN:E/O, and MN/AE alignment result from consistent differences in their respective patterns of co-modulation, or from of greater trial-by-trial variability in the patterns of co-modulation among MNs during observation than during execution, and still greater variability among AE neurons during execution? The bootstrapping approach we used for CCA (see Methods) enabled us to evaluate the consistency of relationships among trajectory segments across repeated samplings of trials recorded from the same neuron population in the same session and in the same context (execution or observation). We therefore performed 500 iterations of CCA between two different random samples of MN execution (MN:E/E), MN observation (MN:O/O), or AE execution (AE:E/E) trajectory segments from a given session (2 R, 3 T, 3 F). This within-group alignment of MN execution trajectory segments from the same session (Figure 8D, MN:E/E, gray, Hold: \begin{document}$\overline{C C 1}=0.88, \overline{C C 2}=0.74, \overline{C C 3}=0.55$\end{document}) was as strong as between session alignment (Figure 8C, MN/1:2, black). But within-group alignment of MN observation trajectory segments (Figure 8D, MN:O/O, orange, Hold: \begin{document}$\overline{C C 1}=0.65, \overline{C C 2}=0.46, \overline{C C 3}=0.24$\end{document}) was lower than that found with MN execution segments Figure 8C, MN:E/O, red, \begin{document}$(\overline{C C 1}=0.73, \overline{C C 2}=0.54, \overline{C C 3}=0.39)$\end{document}. Likewise, within-group alignment of AE neuron trajectory segments (Figure 8D, AE:E/E, light blue, Hold: \begin{document}$\overline{C C 1}=0.46, \overline{C C 2}=0.25, \overline{C C 3}=0.10$\end{document}) was lower than their alignment with MN execution segments (Figure 8C, MN/AE, blue, Hold: \begin{document}$\overline{C C 1}=0.57, \overline{C C 2}=0.35, \overline{C C 3}=0.19$\end{document}). Whereas MN execution trajectories were relatively consistent within sessions, MN observation trajectories and AE execution trajectories were less so.”

And in the Discussion we now suggest (lines 682 to 698):

“Based on the assumption that AE neurons and MNs function as a homogenous neuron population during action execution, we had expected AE and MN execution trajectory segments to align closely. During execution trials, the progression of instantaneous condition-dependent subspaces and of classification accuracy in AE populations was quite similar to that in MN populations. We were surprised to find, therefore, that alignment between execution trajectory segments from AE populations and from the simultaneously recorded MN populations was even lower than alignment between MN execution and observation segments (Figure 8C, blue versus red). Moreover, whereas within-group alignment of MN execution trajectory segments was high, within-group alignment of AE neuron execution trajectory segments was low (Figure 8D, gray versus light blue). These findings indicate that the predominant patterns of co-modulation among MNs during execution are quite consistent within sessions, but the patterns of comodulation among AE neurons are considerably more variable. Together with our previous finding that modulation of MNs leads that of non-mirror neurons in time, both at the single neuron level and at the population level (Mazurek and Schieber, 2019), this difference in consistency versus variability leads us to speculate that during action execution, while MNs carry a consistent forward model of the intended movement, AE neurons carry more variable feedback information.”

(2) The method presented here is similar and perhaps related to principal angles (https://doi.org/10.2307/2005662). It would be interesting to confirm these results with principal angles. For instance, instead of using the decoding performance as a proxy for shifting subspaces, principal angles could directly quantify the 'shift' (similar to Gallego et al, Nat Comm, 2018).

Point taken. We now have calculated the principal angles as a function of time and present them as a new section of the Results including new figure 4 (lines 237 to 293).

“Instantaneous subspaces shift progressively during both execution and observation

We identified an instantaneous subspace at each one millisecond time step of RGM trials. At each time step, we applied PCA to the 4 instantaneous neural states (i.e. the 4 points on the neural trajectories representing trials involving the 4 different objects each averaged across 20 trials per object, totaling 80 trials), yielding a 3-dimensional subspace at that time (see Methods). Note that because these 3-dimensional subspaces are essentially instantaneous, they capture the condition-dependent variation in neural states, but not the common, condition-independent variation. To examine the temporal progression of these instantaneous subspaces, we then calculated the principal angles between each 80-trial instantaneous subspace and the instantaneous subspaces averaged across all trials at four behavioral time points that could be readily defined across trials, sessions, and monkeys: the onset of the instruction (I), the go cue (G), the movement onset (M), and the beginning of the final hold (H). This process was repeated 10 times with replacement to assess the variability of the principal angles. The closer the principal angles are to 0°, the closer the two subspaces are to being identical; the closer to 90°, the closer the two subspaces are to being orthogonal.

Figure 4A-D illustrate the temporal progression of the first principal angle of the mirror neuron population in the three sessions (red, green, and blue) from monkey R during execution trials. As illustrated in Figure 4 – figure supplement 1 (see also the related Methods), in each session all three principal angles, each of which could range from 0° to 90°, tended to follow a similar time course. In the Results we therefore illustrate only the first (i.e. smallest) principal angle. Solid traces represent the mean across 10-fold cross validation using the 80-trial subsets of all the available trials; shading indicates ±1 standard deviation. As would be expected, the instantaneous subspace using 80 trials approaches the subspace using all trials at each of the four selected times—I, G, M, and H—indicated by the relatively narrow trough dipping toward 0°. Of greater interest are the slower changes in the first principal angle in between these four time points. Figure 4A shows that after instruction onset (I) the instantaneous subspace shifted quickly away from the subspace at time I, indicated by a rapid increase in principal angle to levels not much lower than what might be expected by chance alone (horizontal dashed line). In contrast, throughout the remainder of the instruction and delay epochs (from I to G), Figure 4B and C show that the 80-trial instantaneous subspace shifted gradually and concurrently, not sequentially, toward the all-trial subspaces that would be reached at the end of the delay period (G) and then at the onset of movement (M), indicated by the progressive decreases in principal angle. As shown by Figure 4D, shifting toward the H subspace did not begin until the movement onset (M). To summarize, these changes in principal angles indicate that after shifting briefly toward the subspace present at time the instruction appeared (I), the instantaneous subspace shifted progressively throughout the instruction and delay epochs toward the subspace that would be reached at the time of the go cue (G), then further toward that at the time of movement onset (M), and only thereafter shifted toward the instantaneous subspace that would be present at the time of the hold (H).

Figure 4E-H show the progression of the first principal angle of the mirror neuron population during observation trials. Overall, the temporal progression of the MN instantaneous subspace during observation was similar to that found during execution, particularly around times I and H. The decrease in principal angle relative to the G and M instantaneous subspaces during the delay epoch was less pronounced during observation than during execution. Nevertheless, these findings support the hypothesis that the condition-dependent subspace of PM MNs shifts progressively over the time course of RGM trials during both execution and observation, as illustrated schematically in Figure 1A.

We also examined the temporal progression of the instantaneous subspace of AE neurons. As would be expected given that AE neurons were not modulated significantly during observation trials, in the observation context AE populations had no gradual changes in principal angle (Figure 4 – figure supplement 3). During execution, however, Figure 4I-L show that the AE populations had a pattern of gradual decrease in principal angle similar to that found in the MN population (Figure 4A-D). After the instruction onset, the instantaneous subspace shifted quickly away from that present at time I and progressed gradually toward that present at times G and M, only shifting toward that present at time H after movement onset. As for the PM MN populations, the condition-dependent subspace of the PM AE populations shifted progressively over the time course of execution RGM trials.”

The related Methods are now described in subsection “Subspace Comparisons—Principal Angles”

Relatedly, why the decoding of the 'object type' is used to establish the progressive shifting of the subspaces? I would be interested to see the authors' argument.

We have clarified the reason for our decoding analysis as follows (lines 295 to 297):

“The progressive changes in principal angles do not capture another important aspect of condition-dependent neural activity. The neural trajectories during trials involving different objects separated increasingly as trials progressed in time.”

And… (lines 332 to 348):

“Decodable information changes progressively during both execution and observation

As RGM trials proceeded in time, the condition-dependent neural activity of the PM MN population thus changed in two ways. First, the instantaneous condition-dependent subspace shifted, indicating that the patterns of firing-rate co-modulation among neurons representing the four different RGM movements changed progressively, both during execution and during observation. Second, as firing rates generally increased, the neural trajectories representing the four RGM movements became progressively more separated, more so during execution than during observation.

To evaluate the combined effects of these two progressive changes, we clipped 100 ms single-trial trajectory segments beginning at times I, G, M, or H, and projected these trajectory segments from individual trials into the instantaneous 3D subspaces at 50 ms time steps. At each of these time steps, we trained a separate LSTM decoder to classify individual trials according to which of the four objects was involved in that trial. We expected that the trajectory segments would be classified most accurately when projected into instantaneous subspaces near the time at which the trajectory segments were clipped. At other times we reasoned that classification accuracy would depend both on the similarity of the current instantaneous subspace to that found at the clip time as evaluated by the principal angle (Figure 4), and on the separation of the four trajectories at the clip time (Figure 5).”

The object type should be much more decodable during movement or hold, than instruction, which is probably why the chance-level decoding performance (horizontal lines) is twice the instruction segment for the movement segment.

Indeed, the object type is more decodable during the movement and hold than during instruction or delay epochs.

(3) Why aren't execution and observation subspaces compared together directly? Especially given that there are both types of trials in the same session with the same recorded population of neurons. Using instantaneous subspaces, or the principal angles between manifolds during exec trials vs obs trials.

Point taken. We now have added comparison of the execution and observation subspaces using the principal angles between instantaneous subspaces (lines 421 to 436):

“Do PM mirror neurons progress through the same subspaces during execution and observation?

Having found that PM mirror neuron populations show similar progressive shifts in their instantaneous neural subspace during execution and observation of RGM trials, as well as similar changes in decodable information, we then asked whether this progression passes through similar subspaces during execution and observation. To address this question, we first calculated the principal angles between the instantaneous mirror-neuron execution subspace at selected times I, G, M, or H and the entire time series of instantaneous mirror-neuron observation subspaces (Figure 7A-D). Conversely, we calculated the principal angles between the instantaneous observation subspaces at selected times I, G, M, or H and the entire time series of instantaneous execution subspaces (Figure 7E-H). Although the principal angles were slightly smaller than might be expected from chance alone, indicating some minimal overlap of execution and observation instantaneous subspaces, the instantaneous observation subspaces did not show any progressive shift toward the I, G, M, or H execution subspace (Figure 7A-D), nor did the instantaneous execution subspaces shift toward the I, G, M, or H observation subspace (Figure 7E-H).”

(4) The definition of the instantaneous subspaces is a critical point in the manuscript. I think it is slightly unclear: based on the Methods section #715-722 and the main text #173-#181, I gather that the subspaces are based on trial averaged neural activity for each of the 4 objects, separately. So for each object and per timepoint, a vector of size (1, n) -n neurons- is reduced to a vector of (1, 2 or 3 -the main text says 2, methods say 3-) which would be a single point in the low-d space. Is this description accurate? This should be clarified in the manuscript.

In the Methods, we now have clarified (lines 849 to 859):

“Instantaneous subspace identification

Instantaneous neural subspaces were identified at 1 ms intervals. At each 1 ms time step, the N-dimensional neural firing rates from trials involving the four different objects— sphere, button, coaxial cylinder, and perpendicular cylinder—were averaged separately, providing four points in the N-dimensional space representing the average neural activity for trials involving the different objects at that time step. PCA then was performed on these four points. Because three dimensions capture all the variance of four points, three principal component dimensions fully defined each instantaneous subspace. Each instantaneous 3D subspace can be considered a filter described by a matrix, W, that can project high-dimensional neural activity into a low-dimensional subspace, with the time series of instantaneous subspaces, W_i, forming a time series of filters (Figure 1B).”

(5) Isn't the process of projecting segments of neural dynamics and comparing the results equivalent to comparing the projection matrices in the first place? If so, that might have been a more intuitive avenue to follow.

As described in more detail in our responses to item 2, above, we have added analyses of principal angles to compare the projection matrices directly. However, “the process of projecting segments of neural dynamics and comparing the results” incorporates the progressively increasing separation of the trajectory segments and hence is not simply equivalent to comparing the subspaces with principal angles.

(6) Lines #385-#389: This process seems unnecessarily complicated. Also, given the number of trials available, this sometimes doesn't make sense. E.g. Monkey R exec has only 8 trials of one of the objects, so bootstrapping 20 trials 500 times would be spurious. Why not, as per Gallego et al, Nat Neurosci 2020 and Safaie et al, Nat 2023 which are cited, concatenate the trials?

In the Methods we now clarify that (lines 953 to 969):

“To provide an estimate of variability, we used a bootstrapping approach to CCA. From each of two data sets we randomly selected 20 trials involving each target object (totaling 80 trials) with replacement, clipped trajectory segments from each of those trials for 100 ms (100 points at 1 ms intervals) after the instruction onset, go cue, movement onset, or beginning of the final hold, and performed CCA as described above. (Note that because session 1 from monkey R included only 8 button trials (Table 1), we excluded this session from CCA analyses.) With 500 iterations, we obtained a distribution of the correlation coefficients (CCs) between the two data sets in each of the three dimensions of the aligned subspace, which permitted statistical comparisons. We then used this approach to evaluate alignment of latent dynamics between different sessions (e.g. execution trials on two different days), between different contexts (e.g. execution and observation), and between different neural populations (e.g. MNs and AE neurons).This bootstrapping approach further enabled us to assess the consistency of relationships among neural trajectories within a given group—i.e. the same neural population during the same context (execution or observation) in the same session—by drawing two separate random samples of 80 trials from the same population, context, and session (Figure 8D), which would not have been possible had we concatenated trajectory segments from all trials in the session (Gallego et al., 2020; Safaie et al., 2023).”

And we report results that could not have been obtained by concatenating all the trials (lines 522 to 541):

“Did these differences in MN:1/2, MN:E/O, and MN/AE alignment result from consistent differences in their respective patterns of co-modulation, or from of greater trial-by-trial variability in the patterns of co-modulation among MNs during observation than during execution, and still greater variability among AE neurons during execution? The bootstrapping approach we used for CCA (see Methods) enabled us to evaluate the consistency of relationships among trajectory segments across repeated samplings of trials recorded from the same neuron population in the same session and in the same context (execution or observation). We therefore performed 500 iterations of CCA between two different random samples of MN execution (MN:E/E), MN observation (MN:O/O), or AE execution (AE:E/E) trajectory segments from a given session (2 R, 3 T, 3 F). This within-group alignment of MN execution trajectory segments from the same session (Figure 8D, MN:E/E, gray, Hold: \begin{document}$\overline{C C 1}=0.88, \overline{C C 2}=0.74, \overline{C C 3}=0.55$\end{document}) was as strong as between session alignment (Figure 8C, MN/1:2, black). But within-group alignment of MN observation trajectory segments (Figure 8D, MN:O/O, orange, Hold: \begin{document}$\overline{C C 1}=0.65, \overline{C C 2}=0.46, \overline{C C 3}=0.24$\end{document}) was lower than that found with MN execution segments Figure 8C, MN:E/O, red, \begin{document}$(\overline{C C 1}=0.73, \overline{C C 2}=0.54, \overline{C C 3}=0.39)$\end{document}. Likewise, within-group alignment of AE neuron trajectory segments (Figure 8D, AE:E/E, light blue, Hold: \begin{document}$\overline{C C 1}=0.46, \overline{C C 2}=0.25, \overline{C C 3}=0.10$\end{document}) was lower than their alignment with MN execution segments (Figure 8C, MN/AE, blue, Hold: \begin{document}$\overline{C C 1}=0.57, \overline{C C 2}=0.35, \overline{C C 3}=0.19$\end{document}). Whereas MN execution trajectories were relatively consistent within sessions, MN observation trajectories and AE execution trajectories were less so.”

Because only 8 button trials were available in Session 1 from Monkey R, we excluded this session from the CCA analyses. Sessions 2 and 3 from monkey R provide valid results, however. For example, we now state explicitly (lines 468 to 472):

“As a positive control, we first aligned MN execution trajectory segments from two different sessions in the same monkey (which we abbreviate as MN:1/2). The 2 sessions in monkey R provided only 1 possible comparison, but the 3 sessions in monkeys T and F each provided 3 comparisons. For each of these 7 comparisons, we found the bootstrapped average of CC1, of CC2, and of CC3.”

(7) Related to the CCA analysis, what behavioural epoch has been used here, the same as the previous analyses, i.e. 100ms? how many datapoint is that in time? Given that CCA is essentially a correlation value, too few datapoints make it rather meaningless. If that's the case, I encourage using, let's say, one window combined of I and G until movement, and one window of movement and hold, such that they are both easier to interpret. Indeed low values of exec-exec in CC2 compared to Gallego et al, Nat Neurosci, 2020 might be a sign of a methodological error.

In the Methods described for CCA, we now have clarified that (lines 953 to 961):

“To provide an estimate of variability, we used a bootstrapping approach to CCA. From each of two data sets we randomly selected 20 trials involving each target object (totaling 80 trials) with replacement, clipped trajectory segments from each of those trials for 100 ms (100 points at 1 ms intervals) after the instruction onset, go cue, movement onset, or beginning of the final hold, and performed CCA as described above. (Note that because session 1 from monkey R included only 8 button trials (Table 1), we excluded this session from CCA analyses.) With 500 iterations, we obtained a distribution of the correlation coefficients (CCs) between the two data sets in each of the three dimensions of the aligned subspace, which permitted statistical comparisons.”

And in the Results we report that (lines 475 to 480):

“The highest values for MN:1/2 correlations were obtained for the Movement trajectory segments \begin{document}$(\overline{C C 1}=0.89, \overline{C C 2}=0.77, \overline{C C 3}=0.61)$\end{document}. These values indicate consistent relationships among the Movement neural trajectory segments representing the four different RGM movements from session to session, as would have been expected from previous studies (Gallego et al., 2018; Gallego et al., 2020; Safaie et al., 2023).”

**Reviewer #3 (Public Review):**
Summary:In their study, Zhao et al. investigated the population activity of mirror neurons (MNs) in the premotor cortex of monkeys either executing or observing a task consisting of reaching to, grasping, and manipulating various objects. The authors proposed an innovative method for analyzing the population activity of MNs during both execution and observation trials. This method enabled to isolate the condition-dependent variance in neural data and to study its temporal evolution over the course of single trials. The method proposed by the authors consists of building a time series of "instantaneous" subspaces with single time step resolution, rather than a single subspace spanning the entire task duration. As these subspaces are computed on an instant time basis, projecting neural activity from a given task time into them results in latent trajectories that capture condition-dependent variance while minimizing the condition-independent one. The authors then analyzed the time evolution of these instantaneous subspaces and revealed that a progressive shift is present in subspaces of both execution and observation trials, with slower shifts during the grasping and manipulating phases compared to the initial preparation phase. Finally, they compared the instantaneous subspaces between execution and observation trials and observed that neural population activity did not traverse the same subspaces in these two conditions. However, they showed that these distinct neural representations can be aligned with Canonical Correlation Analysis, indicating dynamic similarities of neural data when executing and observing the task. The authors speculated that such similarities might facilitate the nervous system's ability to recognize actions performed by oneself or another individual.Strengths:Unlike other areas of the brain, the analysis of neural population dynamics of premotor cortex MNs is not well established. Furthermore, analyzing population activity recorded during non-trivial motor actions, distinct from the commonly used reaching tasks, serves as a valuable contribution to computational neuroscience. This study holds particular significance as it bridges both domains, shedding light on the temporal evolution of the shift in neural states when executing and observing actions. The results are moderately robust, and the proposed analytical method could potentially be used in other neuroscience contexts.Weaknesses:While the overall clarity is satisfactory, the paper falls short in providing a clear description of the mathematical formulas for the different methods used in the study.

We have added the various mathematical formulas in the Methods.

For Cumulative Separation (lines 864 to 871):

“To quantify the separation between the four trial-averaged trajectory segments involving the different objects in a given instantaneous subspace, we then calculated their cumulative separation (𝐶𝑆) as:\begin{document}$$\displaystyle  C S=\frac{1}{T} \sum_{t \in T} D(t)=\frac{1}{T} \sum_{t \in T} \sum_{i \neq j} d_{i j}(t)$$\end{document}

where *dij(t)* is the 3-dimensional Euclidean distance between the *ith* and *jth* trajectories at time point 𝑡. We summed the 6 pairwise distances between the 4 trajectory segments across time points and normalized by the number of time points, 𝑇 = 100. The larger the 𝐶𝑆, the greater the separation of the trajectory segments.”

For principal angles (lines 877 to 884):

**“**For example, given the 3-dimensional instantaneous subspace at the time of movement onset, *WM* and at any other time, *Wi*, we calculated their 3x3 inner product matrix and performed singular value decomposition to obtain:\begin{document}$$\displaystyle W_{M}^{T} W_{i}=P_{M} C P_{i}^{T}$$\end{document}

where 3x3 matrices *PM* and *WP* define new manifold directions which successively minimize the 3 principal angles specific to the two subspaces being compared. The elements of diagonal matrix 𝐶 then are the ranked cosines of the principal angles, 𝜃𝑖 , ordered from smallest to largest:\begin{document}$$\displaystyle C=\operatorname{diag}\left(\cos \left(\theta_{1}\right), \cos \left(\theta_{2}\right), \cos \left(\theta_{3}\right)\right)$$\end{document}

For CCA (lines 945 to 952):

“CCA was performed as follows: The original latent dynamics, *LA* and *LB*, first were transformed and decomposed as \begin{document}$L_{A}^{T}=Q_{A} R_{A}$\end{document} and \begin{document}$L_{B}^{T}=Q_{B} R_{B}$\end{document}. The first m = 3 column vectors of each 𝑄𝑖 provide an orthonormal basis for the column vectors of \begin{document}$L_{i}^{T}$\end{document} (where 𝑖 = 𝐴, 𝐵). Singular value decomposition on the inner product matrix of 𝑄𝐴 and 𝑄𝐵 then gives \begin{document}$Q_{A}^{T} Q_{B}=U S V^{T}$\end{document}, and new manifold directions that maximize pairwise correlations are provided by \begin{document}$M_{A}=R_{A}^{-1} U$\end{document} and \begin{document}$M_{R}=R_{R}^{-1} V$\end{document}. We then projected the original latent dynamics into the new, common subspace: \begin{document}$\tilde{L}_{A}^{T}=L_{A}^{T} M_{A} ; \quad \tilde{L}_{B}^{T}=L_{B}^{T} M_{B}$\end{document}. Pairwise correlation coefficients between the aligned latent dynamics sorted from largest to smallest then are given by the elements of the diagonal matrix \begin{document}$S=\tilde{L}_{A} \tilde{L}_{B}^{T}$\end{document}.”

Moreover, it was not immediately clear why the authors did not consider a (relatively) straightforward metric to quantity the progressive shift of the instantaneous subspaces, such as computing the angle between consecutive subspaces, rather than choosing a (in my opinion) more cumbersome metric based on classification of trajectory segments representing different movements.

Point taken. We now have calculated the principal angles as a function of time and present them as a new section of the Results including new figure 4 (lines 237 to 293).

“Instantaneous subspaces shift progressively during both execution and observation

We identified an instantaneous subspace at each one millisecond time step of RGM trials. At each time step, we applied PCA to the 4 instantaneous neural states (i.e. the 4 points on the neural trajectories representing trials involving the 4 different objects each averaged across 20 trials per object, totaling 80 trials), yielding a 3-dimensional subspace at that time (see Methods). Note that because these 3-dimensional subspaces are essentially instantaneous, they capture the condition-dependent variation in neural states, but not the common, condition-independent variation. To examine the temporal progression of these instantaneous subspaces, we then calculated the principal angles between each 80-trial instantaneous subspace and the instantaneous subspaces averaged across all trials at four behavioral time points that could be readily defined across trials, sessions, and monkeys: the onset of the instruction (I), the go cue (G), the movement onset (M), and the beginning of the final hold (H). This process was repeated 10 times with replacement to assess the variability of the principal angles. The closer the principal angles are to 0°, the closer the two subspaces are to being identical; the closer to 90°, the closer the two subspaces are to being orthogonal.

Figure 4A-D illustrate the temporal progression of the first principal angle of the mirror neuron population in the three sessions (red, green, and blue) from monkey R during execution trials. As illustrated in Figure 4 – figure supplement 1 (see also the related Methods), in each session all three principal angles, each of which could range from 0° to 90°, tended to follow a similar time course. In the Results we therefore illustrate only the first (i.e. smallest) principal angle. Solid traces represent the mean across 10-fold cross validation using the 80-trial subsets of all the available trials; shading indicates ±1 standard deviation. As would be expected, the instantaneous subspace using 80 trials approaches the subspace using all trials at each of the four selected times—I, G, M, and H—indicated by the relatively narrow trough dipping toward 0°. Of greater interest are the slower changes in the first principal angle in between these four time points. Figure 4A shows that after instruction onset (I) the instantaneous subspace shifted quickly away from the subspace at time I, indicated by a rapid increase in principal angle to levels not much lower than what might be expected by chance alone (horizontal dashed line). In contrast, throughout the remainder of the instruction and delay epochs (from I to G), Figure 4B and C show that the 80-trial instantaneous subspace shifted gradually and concurrently, not sequentially, toward the all-trial subspaces that would be reached at the end of the delay period (G) and then at the onset of movement (M), indicated by the progressive decreases in principal angle. As shown by Figure 4D, shifting toward the H subspace did not begin until the movement onset (M). To summarize, these changes in principal angles indicate that after shifting briefly toward the subspace present at time the instruction appeared (I), the instantaneous subspace shifted progressively throughout the instruction and delay epochs toward the subspace that would be reached at the time of the go cue (G), then further toward that at the time of movement onset (M), and only thereafter shifted toward the instantaneous subspace that would be present at the time of the hold (H).

Figure 4E-H show the progression of the first principal angle of the mirror neuron population during observation trials. Overall, the temporal progression of the MN instantaneous subspace during observation was similar to that found during execution, particularly around times I and H. The decrease in principal angle relative to the G and M instantaneous subspaces during the delay epoch was less pronounced during observation than during execution. Nevertheless, these findings support the hypothesis that the condition-dependent subspace of PM MNs shifts progressively over the time course of RGM trials during both execution and observation, as illustrated schematically in Figure 1A.

We also examined the temporal progression of the instantaneous subspace of AE neurons. As would be expected given that AE neurons were not modulated significantly during observation trials, in the observation context AE populations had no gradual changes in principal angle (Figure 4 – figure supplement 3). During execution, however, Figure 4I-L show that the AE populations had a pattern of gradual decrease in principal angle similar to that found in the MN population (Figure 4A-D). After the instruction onset, the instantaneous subspace shifted quickly away from that present at time I and progressed gradually toward that present at times G and M, only shifting toward that present at time H after movement onset. As for the PM MN populations, the condition-dependent subspace of the PM AE populations shifted progressively over the time course of execution RGM trials.”

The related Methods are now described in subsection “Subspace Comparisons—Principal Angles”

Specific comments:In the methods, it is stated that instantaneous subspaces are found with 3 PCs. Why does it say 2 here?

We now have clarified. (lines 295 to 310):

“The progressive changes in principal angles do not capture another important aspect of condition-dependent neural activity. The neural trajectories during trials involving different objects separated increasingly as trials progressed in time. To illustrate this increasing separation, we clipped 100 ms segments of high-dimensional MN population trial-averaged trajectories beginning at times I, G, M, and H, for trials involving each of the four objects. We then projected the set of four object-specific trajectory segments clipped at each time into each of the four instantaneous 3D subspaces at times I, G, M, and H. This process was repeated separately for execution trials and for observation trials.

For visualization, we projected these trial-averaged trajectory segments from an example session into the PC1 vs PC2 planes (which consistently captured > 70% of the variance) of the I, G, M, or H instantaneous 3D subspaces. In Figure 5, the trajectory segments for each of the four objects (sphere – purple, button – cyan, coaxial cylinder – magenta, perpendicular cylinder – yellow) sampled at different times (rows) have been projected into each of the four instantaneous subspaces defined at different times (columns). Rather than appearing knotted as in Figure 3, these short trajectory segments are distinct when projected into each instantaneous subspace.”

And in the legend for Figure 5 we now clarify that:

“Each set of these four segments then was projected into the PC1 vs PC2 plane of the instantaneous 3D subspace present at four different times (columns: I, G, M, H).”

Another doubt on how instantaneous subspaces are computed: in the methods you state that you apply PCA on trial-averaged activity at each 50ms time step. From the next sentence, I gather that you apply PCA on an Nx4 data matrix (N being the number of neurons, and 4 being the trial-averaged activity of the four objects) every 50 ms. Is this right? It would help to explicitly specify the dimensions of the data matrix that goes into PCA computation.

We apologize for this confusion. Although the LSTM decoding was performed in 50 ms time steps, the instantaneous subspaces were calculated at 1 ms intervals. In the Methods we now have clarified (lines 849 to 759):

“Instantaneous subspace identification

Instantaneous neural subspaces were identified at 1 ms intervals. At each 1 ms time step, the N-dimensional neural firing rates from trials involving the four different objects— sphere, button, coaxial cylinder, and perpendicular cylinder—were averaged separately, providing four points in the N-dimensional space representing the average neural activity for trials involving the different objects at that time step. PCA then was performed on these four points. Because three dimensions capture all the variance of four points, three principal component dimensions fully defined each instantaneous subspace. Each instantaneous 3D subspace can be considered a filter described by a matrix, W, that can project high-dimensional neural activity into a low-dimensional subspace, with the time series of instantaneous subspaces, W_i, forming a time series of filters (Figure 1B).”

It would help to include some equations in the methods section related to the LSTM decoding. Just to make sure I understood correctly: after having identified the instantaneous subspaces (every 50 ms), you projected the Instruction, Go, Movement, and Holding segments from individual trials (each containing 100 samples, since they are sampled from a 100ms window) onto each instantaneous subspace. So you have four trajectories for each subspace. In the methods, it is stated that a single LSTM classifier is trained for each subspace. Do you also have a separate classifier for each trajectory segment? What is used as input to the classifier? Each trajectory segment should be a 100x3 matrix once projected in an instantaneous subspace. Is that what (each of) the LSTMs take as input? And lastly, what is the LSTM trained to predict exactly? Just a label indicating the type of object that was manipulated in that trial? I apologize if I overlooked any detail, but I believe a clearer explanation of the LSTM, preferably with mathematical formulas, would greatly help readers understand this section.

LSTM decoding is not readily described with a set of equations. However, we have expanded our description to provide the information requested (lines 910 to 937):

“Decodable information—LSTM

As illustrated schematically in Figure 1B, the same segment of high-dimensional neural activity projected into different instantaneous subspaces can generate low-dimensional trajectories of varying separation. The degree of separation among the projected trajectory segments will depend, not only on their separation at the time when the segments were clipped, but also on the similarity of the subspaces into which the trajectory segments are projected. To quantify the combined effects of trajectory separation and projection into different subspaces, we projected high-dimensional neural trajectory segments (each including 100 points at 1 ms intervals) from successful trials involving each of the four different target objects into time series of 3-dimensional instantaneous subspaces at 50 ms intervals. In each of these instantaneous subspaces, the neural trajectory segment from each trial thus became a 100 point x 3 dimensional matrix. For each instantaneous subspace in the time series, we then trained a separate long short-term memory (LSTM, (Hochreiter and Schmidhuber, 1997)) classifier to attribute each of the neural trajectories from individual trials to one of the four target object labels: sphere, button, coaxial cylinder, or perpendicular cylinder. Using MATLAB’s Deep Learning Toolbox, each LSTM classifier had 3 inputs (instantaneous subspace dimensions), 20 hidden units in the bidirectional LSTM layer, and a softmax layer preceding the classification layer which had 4 output classes (target objects). The total number of successful trials available in each session for each object is given in Table 1. To avoid bias based on the total number of successful trials, we used the minimum number of successful trials across the four objects in each session, selecting that number from the total available randomly with replacement. Each LSTM classifier was trained with MATLAB’s adaptive moment estimation (Adam) optimizer on 40% of the selected trials, and the remaining 60% were decoded by the trained classifier. The success of this decoding was used as an estimate of classification accuracy from 0 (no correct classifications) to 1 (100% correct classifications). This process was repeated 10 times and the mean ± standard deviation across the 10 folds was reported as the classification accuracy at that time. Classification accuracy of trials projected into each instantaneous subspace at 50 ms intervals was plotted as a function of trial time.”

**Recommendations for the authors:**

**Reviewer #1 (Recommendations For The Authors):**
Here are some more specific comments.Abstract. Line 41. "same action" is not justified, there is plenty of evidence showing that the action does not need to be the same (or it has not even to be an action), rephrasing or substituting with "similar" is necessary, especially in the light of the subsequent sentence (which is totally correct).

Thank you for pointing this out. As recommended, we have changed “same” to “similar” (lines 40 to 41):

“Many neurons in the premotor cortex show firing rate modulation whether the subject performs an action or observes another individual performing a similar action.”

Introduction. A relevant, missing reference in the otherwise exhaustive introduction is Albertini et al. 2021 J Neurophysiol, showing that neural dynamics and similarities between biological and nonbiological movements in premotor areas are greater than those between the same executed and observed movements.

Thank you for pointing out this important finding. After revision, we felt it was now cited most appropriately in the revised Discussion as follows (lines 730 to 736):

“Alternatively, given that observation of another individual can be considered a form of social interaction, PM MN population activity during action observation, rather than representing movements made by another individual similar to one’s own movements, instead may represent different movements one might execute oneself in response to those made by another individual (Ninomiya et al., 2020; Bonini et al., 2022; Ferrucci et al., 2022; Pomper et al., 2023). This possibility is consistent with the finding that the neural dynamics of PM MN populations are more similar during observation of biological versus non-biological movements than during execution versus observation (Albertini et al., 2021)."

In Line 85, the sentence about Papadourakis and Raos 2019 has to be generalized to PMv, as they show that the proportion of congruent MNs is at chance in both PMd and PMv.

Point taken. We have rephrased this sentence as follows (lines 88 to 89):

“And in both PMv and PMd, the proportion of congruent neurons may not be different from that expected by chance alone (Papadourakis and Raos, 2019).”

Lines 122-132. The initial sentence was unclear to me at first glance. I was wondering how subspaces could be "at other times over the course of the trial" if they are instantaneous. I could imagine that the subspaces referred to corresponding behavioral intervals of execution and observation conditions (and this may be what they will later call "condition dependent" activity), but nevertheless, they could hardly be understood as "instantaneous". I grasped the author's idea only when reading the results, with the statement "no-time dependent variance is captured". The idea is to take a static snapshot of the evolution of population activity at each checkpoint (i.e. I, G, M, and H): I suggest clarifying this point immediately in the introduction to improve readability.

We have clarified this point by adding two paragraphs to the Introduction first defining condition independent versus condition-dependent variance and then explaining the use of instantaneous subspaces (lines 125 to 153):

“A relevant but often overlooked aspect of such dynamics in neuron populations active during both execution and observation has to do with the distinction between condition independent and condition-dependent variation in neuronal activity (Kaufman et al., 2016; Rouse and Schieber, 2018). The variance in neural activity averaged across all the conditions in a given task context is condition-independent. For example, in an 8-direction center-out reaching task, averaging a unit’s firing rate as a function of time across all 8 directions may show an initially low firing rate that increases prior to movement onset, peaks during the movement, and then declines during the final hold, irrespective of the movement direction. Subtracting this condition-independent activity from the unit’s firing rate during each trial gives the remaining variance, and averaging separately across trials in each of the 8 directions then averages out noise variance, leaving the condition-dependent variance that represents the unit’s modulation among the 8 directions (conditions). Alternatively, condition-independent, condition dependent, and noise variance can be partitioned through demixed principal component analysis (Kobak et al., 2016; Gallego et al., 2018). The extent to which neural dynamics occur in a subspace shared by execution and observation versus subspaces unique to execution or observation may differ for the condition-independent versus condition-dependent partitions of neural activity. Here, we tested the hypothesis that the condition-dependent activity of PM mirror neuron populations progresses through distinct subspaces during execution versus observation, which would indicate distinct patterns of co-modulation amongst mirror neurons during execution versus observation.

Because of the complexity of condition-dependent neural trajectories for movements involving the hand, we developed a novel approach. Rather than examining trajectories over the entire time course of behavioral trials, we identified time series of instantaneous PM mirror neuron subspaces covering the time course of behavioral trials. We identified separate time series for execution trials and for observation trials, both involving four different reach-graspmanipulation (RGM) movements. Given that each subspace in these time series is instantaneous (a snapshot in time), it captures condition-dependent variance in the neural activity among the four RGM movements while minimizing condition-independent (time dependent) variance.”

Results.Regarding the execution-observation alignment, as explained in my initial comment, it does not sound convincing. Applying a CCA to align EXE and OBS activities (which the authors had just shown being essentially not aligned), even separately for each epoch segment (line 396), seems to be a trick to show that they nonetheless share some similarities. Couldn't this be applied to any pairs of differently encoded conditions to create some sort of artificial link between them? Is the similarity in the neural data or rather in the method used to realign them?

CCA would not align arbitrary sets of neural data. The similarity is in the data, not in the method. For example, in an 8-direction center-out task, the neural representation of movement to the 45° target is between the neural representations of the 0° and the 90° targets. If the same is true in a second data set, then CCA will give high correlation coefficients. But if in the second data set the neural representation of the 45° target is between the 135° and 180° targets, CCA will give low correlation coefficients.

In the end, what does this tell us about the brain?

In the Introduction we now clarify that (lines 166 to 170):

“Such alignment would indicate that the relationships among the trajectory segments in the execution subspace are similar to the relationships among the trajectory segments in the observation subspace, indicating a corresponding structure in the latent dynamic representations of execution and observation movements by the same PM MN population.”

And in the Results (lines 449 to 455):

“For example, the trajectories of PMd+M1 neuron populations recorded from two different monkeys during center-out reaching movements could be aligned well (Safaie et al., 2023). CCA showed, for example, that in both brains the neural trajectory for the movement to the target at 0° was closer to the trajectory for movement to the target at 45° than to the trajectory for the movement to the target at 180°. Relationships among these latent dynamic representations of the eight movements thus were similar even though the neural populations were recorded from two different monkeys.”

In relation to Figure 8 (lines 461 to 467)

“But when both sets of trajectory segments are projected into another common subspace identified with CCA, as shown in Figure 8B, a similar relationship among the neural representations of the four movements during execution and observation is revealed. In both behavioral contexts the neural representation of movements involving the sphere (purple) is now closest to the representation of movements involving the coaxial cylinder (magenta) and farthest from that of movements involving the button (cyan). The two sets of trajectory segments are more or less “aligned.”

And in the Discussion (lines 665 to 674):

“Corresponding neural representations of action execution and observation during task epochs with higher neural firing rates have been described previously in PMd MNs and in PMv MNs using representational similarity analysis RSA (Papadourakis and Raos, 2019). And during force production in eight different directions, neural trajectories of PMd neurons draw similar “clocks” during execution, cooperative execution, and passive observation (Pezzulo et al., 2022). Likewise in the present study, despite execution and observation trajectories progressing through largely distinct subspaces, in all three monkeys execution and observation trajectory segments showed some degree of alignment, particularly the Movement and Hold segments (Figure 12A), indicating similar relationships among the latent dynamic representations of the four RGM movements during execution and observation.”

Concerning the discussion, I would like to reconsider it after having seen the authors' response to the comments above and to my general concern about the relevance of the findings from the neurophysiological point of view.

Certainly, please do.

**Reviewer #2 (Recommendations For The Authors):**
Here are a few issues that I want to bring to the authors' attention (in no particular order):• I am not clear on what is meant by "condition-dependent". Is the condition exec vs obs, or the object types?

In the Introduction, we now clarify (lines 125 to 144):

“A relevant but often overlooked aspect of such dynamics in neuron populations active during both execution and observation has to do with the distinction between condition independent and condition-dependent variation in neuronal activity (Kaufman et al., 2016; Rouse and Schieber, 2018). The variance in neural activity averaged across all the conditions in a given task context is condition-independent. For example, in an 8-direction center-out reaching task, averaging a unit’s firing rate as a function of time across all 8 directions may show an initially low firing rate that increases prior to movement onset, peaks during the movement, and then declines during the final hold, irrespective of the movement direction. Subtracting this condition-independent activity from the unit’s firing rate during each trial gives the remaining variance, and averaging separately across trials in each of the 8 directions then averages out noise variance, leaving the condition-dependent variance that represents the unit’s modulation among the 8 directions (conditions). Alternatively, condition-independent, condition dependent, and noise variance can be partitioned through demixed principal component analysis (Kobak et al., 2016; Gallego et al., 2018). The extent to which neural dynamics occur in a subspace shared by execution and observation versus subspaces unique to execution or observation may differ for the condition-independent versus condition-dependent partitions of neural activity. Here, we tested the hypothesis that the condition-dependent activity of PM mirror neuron populations progresses through distinct subspaces during execution versus observation, which would indicate distinct patterns of co-modulation amongst mirror neurons during execution versus observation.”

And in the Results, we have added a new Figure 3 to illustrate condition-independent versus conditiondependent activity using an example from the present data sets (lines 208 to 236):

“Condition-dependent versus condition-independent neural activity in PM MNs

Whereas a large fraction of condition-dependent neural variance during reaching movements without grasping can be captured in a two-dimensional subspace (Churchland et al., 2012; Ames et al., 2014), condition-dependent activity in movements that involve grasping is more complex (Suresh et al., 2020). In part, this may reflect the greater complexity of controlling the 24 degrees of freedom in the hand and wrist as compared to the 4 degrees of freedom in the elbow and shoulder (Sobinov and Bensmaia, 2021). Figure 3 illustrates this complexity in a PM MN population during the present RGM movements. Here, PCA was performed on the activity of a PM MN population across the entire time course of execution trials involving all four objects. The colored traces in Figure 3A show neural trajectories averaged separately across trials involving each of the four objects and then projected into the PC1 vs PC2 plane of the total neural space. Most of the variance in these four trajectories is comprised of a shared rotational component. The black trajectory, obtained by averaging trajectories from trials involving all four objects together, represents this condition-independent (i.e. independent of the object involved) activity. The condition-dependent (i.e. dependent on which object was involved) variation in activity is reflected by the variation in the colored trajectories around the black trajectory. The condition-dependent portions can be isolated by subtracting the black trajectory from each of the colored trajectories. The resulting four condition dependent trajectories have been projected into the PC1 vs PC2 plane of their own common subspace in Figure 3B. Rather than exhibiting a simple rotational motif, these trajectories appear knotted. To better understand how these complex, condition-dependent trajectories progress over the time course of RGM trials, we chose to examine time series of instantaneous subspaces.”

While there is an emphasis on the higher complexity of manipulating objects compared to just reaching movements in the Abstract, the majority of the analysis relates to the instruction, movement initiation, and grasp, and there is no specific analyses looking at manipulation and how those presumably more complex dynamics compare to the reaching dynamics, and how they differ from reaching in the mirror neurons.

We have clarified that (lines 178 to 187):

“Because we chose to study relatively naturalistic movements, the reach, grasp, and manipulation components were not performed separately, but rather in a continuous fluid motion during the movement epoch of the task sequence (Figure 2B). In previous studies involving a version of this task without separate instruction and delay epochs, we have shown that joint kinematics, EMG activity, and neuron activity in the primary motor cortex, all vary throughout the movement epoch in relation to both reach location and object grasped, with location predominating early in the movement epoch and object predominating later (Rouse and Schieber, 2015, 2016a, b). The present task, however, did not dissociate the reach, the hand shape used to grasp the object, and the manipulation performed on the object.”

• The analysis in Fig3C,D is interesting, however, in my opinion, requires control. For instance, what would these values look like if you projected the segments to a subspace defined by the activity during the entire length of the trial, or if you projected the activity during intertrials, just to get a sense of how meaningful these values are?

This material is now presented in Figure 5 – figure supplement 1. In the legend to this figure supplement, we have clarified that (lines 327 to 328):

“CS values, which we use only to characterize the phenomenon of trajectory separation,….”

• MN is used (#85) before definition (#91). Similar for RGM, I believe.

Thanks for catching this problem. We have now defined these abbreviations at first use as follows:

In lines 89 to 92:

“Though many authors apply the term mirror neurons strictly to highly congruent neurons, here we will refer to all neurons modulated during both contexts—execution and observation—as mirror neurons (MNs).”

And in lines 148 to 150:

We identified separate time series for execution trials and for observation trials, both involving four different reach-grasp-manipulation (RGM) movements.”

• I believe in the Intro when presenting the three hypotheses, there is a First, and a Third, but no Second.

We have revised this part of the Introduction without numbering our hypotheses as follows (lines 145 to 173):

“Because of the complexity of condition-dependent neural trajectories for movements involving the hand, we developed a novel approach. Rather than examining trajectories over the entire time course of behavioral trials, we identified time series of instantaneous PM mirror neuron subspaces covering the time course of behavioral trials. We identified separate time series for execution trials and for observation trials, both involving four different reach-graspmanipulation (RGM) movements. Given that each subspace in these time series is instantaneous (a snapshot in time), it captures condition-dependent variance in the neural activity among the four RGM movements while minimizing condition-independent (time dependent) variance.

We then tested the hypothesis that the condition-dependent subspace shifts progressively over the time course of behavioral trials (Figure 1A) by calculating the principal angles between four selected instantaneous subspaces that occurred at times easily defined in each behavioral trial—instruction onset (I), go cue (G), movement onset (M), and the beginning of the final hold (H)—and every other instantaneous subspace in the time series. Initial analyses showed that condition-dependent neural trajectories for the four RGM movements tended to separate increasingly over the course of behavioral trials. We therefore additionally examined the combined effects of (i) the progressively shifting subspaces and (ii) the increasing trajectory separation, by decoding neural trajectory segments sampled for 100 msec after times I, G, M, and H and projected into the time series of instantaneous subspaces (Figure 1B).

Finally, we used canonical correlation to ask whether the prevalent patterns of mirror neuron co-modulation showed similar relationships among the four RGM movements during execution and observation (Figure 1C). Such alignment would indicate that the relationships among the trajectory segments in the execution subspace are similar to the relationships among the trajectory segments in the observation subspace, indicating a corresponding structure in the latent dynamic representations of execution and observation movements by the same PM MN population. And finally, because we previously have found that during action execution the activity of PM mirror neurons tends to lead that of non-mirror neurons which are active only during action execution (AE neurons) (Mazurek and Schieber, 2019), we performed parallel analyses of the instantaneous state space of PM AE neurons.”

• The use of the term 'instantaneous subspaces' in the abstract confused me initially, as I wasn't sure what it meant. It might be a good idea to define or rephrase it.

In the Abstract we now state (lines 51 to 52):

“Rather than following neural trajectories in subspaces that contain their entire time course, we identified time series of instantaneous subspaces …”

And in the Introduction, we have clarified (lines 145 to 153):

“Because of the complexity of condition-dependent neural trajectories for movements involving the hand, we developed a novel approach. Rather than examining trajectories over the entire time course of behavioral trials, we identified time series of instantaneous PM mirror neuron subspaces covering the time course of behavioral trials. We identified separate time series for execution trials and for observation trials, both involving four different reach-graspmanipulation (RGM) movements. Given that each subspace in these time series is instantaneous (a snapshot in time), it captures condition-dependent variance in the neural activity among the four RGM movements while minimizing condition-independent (time dependent) variance.”

And in the Methods (lines 849 to 859):

“Instantaneous subspace identification

Instantaneous neural subspaces were identified at 1 ms intervals. At each 1 ms time step, the N-dimensional neural firing rates from trials involving the four different objects— sphere, button, coaxial cylinder, and perpendicular cylinder—were averaged separately, providing four points in the N-dimensional space representing the average neural activity for trials involving the different objects at that time step. PCA then was performed on these four points. Because three dimensions capture all the variance of four points, three principal component dimensions fully defined each instantaneous subspace. Each instantaneous 3D subspace can be considered a filter described by a matrix, 𝑊, that can project high-dimensional neural activity into a low-dimensional subspace, with the time series of instantaneous subspaces, 𝑊𝑖, forming a time series of filters (Figure 1B).”

**Reviewer #3 (Recommendations For The Authors):**
(1) Page 4, lines 127-131. In the introduction, it was not immediately clear to me what you meant by 'separation' and 'decoding' of the projected neural activity. You do mention that you are separating/decoding trajectory segments representing different movements at the end of this paragraph, but at this point of the paper it was not very clear to me what those different movements were (I only understood that after reading the results section). I suggest briefly expanding on these concepts here.

To clarify these points in the Introduction, we have expanded exposition of these concepts (lines 145 to 163):

“Because of the complexity of condition-dependent neural trajectories for movements involving the hand, we developed a novel approach. Rather than examining trajectories over the entire time course of behavioral trials, we identified time series of instantaneous PM mirror neuron subspaces covering the time course of behavioral trials. We identified separate time series for execution trials and for observation trials, both involving four different reach-graspmanipulation (RGM) movements. Given that each subspace in these time series is instantaneous (a snapshot in time), it captures condition-dependent variance in the neural activity among the four RGM movements while minimizing condition-independent (time dependent) variance.

We then tested the hypothesis that the condition-dependent subspace shifts progressively over the time course of behavioral trials (Figure 1A) by calculating the principal angles between four selected instantaneous subspaces that occurred at times easily defined in each behavioral trial—instruction onset (I), go cue (G), movement onset (M), and the beginning of the final hold (H)—and every other instantaneous subspace in the time series. Initial analyses showed that condition-dependent neural trajectories for the four RGM movements tended to separate increasingly over the course of behavioral trials. We therefore additionally examined the combined effects of (i) the progressively shifting subspaces and (ii) the increasing trajectory separation, by decoding neural trajectory segments sampled for 100 msec after times I, G, M, and H and projected into the time series of instantaneous subspaces (Figure 1B).”

(2) Page 6, line 175. In the methods, it is stated that instantaneous subspaces are found with 3 PCs. Why does it say 2 here?

Thank you for noticing this discrepancy. In the Methods, we have clarified that the instantaneous subspaces are 3-dimensional (see our reply to the next comment), but in Figure 5 (previously Figure 3), for purposes of visualization, we are projecting trajectory segments into the PC1-PC2 plane (lines 295 to 308):

“The progressive changes in principal angles do not capture another important aspect of condition-dependent neural activity. The neural trajectories during trials involving different objects separated increasingly as trials progressed in time. To illustrate this increasing separation, we clipped 100 ms segments of high-dimensional MN population trial-averaged trajectories beginning at times I, G, M, and H, for trials involving each of the four objects. We then projected the set of four object-specific trajectory segments clipped at each time into each of the four instantaneous 3D subspaces at times I, G, M, and H. This process was repeated separately for execution trials and for observation trials.

For visualization, we projected these trial-averaged trajectory segments from an example session into the PC1 vs PC2 planes (which consistently captured > 70% of the variance) of the I, G, M, or H instantaneous 3D subspaces. In Figure 5, the trajectory segments for each of the four objects (sphere – purple, button – cyan, coaxial cylinder – magenta, perpendicular cylinder – yellow) sampled at different times (rows) have been projected into each of the four instantaneous subspaces defined at different times (columns).”

And in the legend for Figure 5 we now clarify that:

“Each set of these four segments then was projected into the PC1 vs PC2 plane of the instantaneous 3D subspace present at four different times (columns: I, G, M, H).”

Another doubt on how instantaneous subspaces are computed: in the methods you state that you apply PCA on trial-averaged activity at each 50ms time step. From the next sentence, I gather that you apply PCA on an Nx4 data matrix (N being the number of neurons, and 4 being the trial-averaged activity of the four objects) every 50 ms. Is this right? It would help to explicitly specify the dimensions of the data matrix that goes into PCA computation.

Thank you for catching an error: The instantaneous subspaces were computed at 1 ms intervals. (It is the LSTM decoding that was done in 50 ms time steps). We have clarified how the instantaneous subspaces were computed in the Methods (lines 849 to 859):

“Instantaneous subspace identification

Instantaneous neural subspaces were identified at 1 ms intervals. At each 1 ms time step, the N-dimensional neural firing rates from trials involving the four different objects— sphere, button, coaxial cylinder, and perpendicular cylinder—were averaged separately, providing four points in the N-dimensional space representing the average neural activity for trials involving the different objects at that time step. PCA then was performed on these four points. Because three dimensions capture all the variance of four points, three principal component dimensions fully defined each instantaneous subspace. Each instantaneous 3D subspace can be considered a filter described by a matrix, 𝑊, that can project high-dimensional neural activity into a low-dimensional subspace, with the time series of instantaneous subspaces, 𝑊𝑖, forming a time series of filters (Figure 1B).”

(3) Page 7, line 210-212. I am not sure if I missed it in the discussion, but have you speculated on why the greatest separation in observation trials was observed during the holding phase while in execution trials during the movement phase?

This was a consistent finding, and we therefore point it out as a difference between execution and observation. Of course, this reflects greater condition-dependent variance in the PM MN population in the movement epoch than in the hold epoch during execution, whereas the reverse is true during observation. We have no clear speculation as to why this occurs, however.

(4) Figure 3. Add a legend with color scheme for each object in panels A and B. Also, please specify what metric is represented by the colorbar of panels C, D, E, F (write it down next to the colorbar itself and not just in the caption).

This is now Figure 5. We have added a color legend for A and B. Panels C, D, E, and F, now have been moved to Figure 5 – figure supplement 1, where we have indicated that the colorbar represents cumulative separation.

(5) Page 9, line 228. I found the description of this decoding analysis a bit confusing initially (and perhaps still do), this should be clarified.

We have clarified our decoding analysis in the Methods (lines 910 to 937):

“Decodable information—LSTM

As illustrated schematically in Figure 1B, the same segment of high-dimensional neural activity projected into different instantaneous subspaces can generate low-dimensional trajectories of varying separation. The degree of separation among the projected trajectory segments will depend, not only on their separation at the time when the segments were clipped, but also on the similarity of the subspaces into which the trajectory segments are projected. To quantify the combined effects of trajectory separation and projection into different subspaces, we projected high-dimensional neural trajectory segments (each including 100 points at 1 ms intervals) from successful trials involving each of the four different target objects into time series of 3-dimensional instantaneous subspaces at 50 ms intervals. In each of these instantaneous subspaces, the neural trajectory segment from each trial thus became a 100 point x 3 dimensional matrix. For each instantaneous subspace in the time series, we then trained a separate long short-term memory (LSTM, (Hochreiter and Schmidhuber, 1997)) classifier to attribute each of the neural trajectories from individual trials to one of the four target object labels: sphere, button, coaxial cylinder, or perpendicular cylinder. Using MATLAB’s Deep Learning Toolbox, each LSTM classifier had 3 inputs (instantaneous subspace dimensions), 20 hidden units in the bidirectional LSTM layer, and a softmax layer preceding the classification layer which had 4 output classes (target objects). The total number of successful trials available in each session for each object is given in Table 1. To avoid bias based on the total number of successful trials, we used the minimum number of successful trials across the four objects in each session, selecting that number from the total available randomly with replacement. Each LSTM classifier was trained with MATLAB’s adaptive moment estimation (Adam) optimizer on 40% of the selected trials, and the remaining 60% were decoded by the trained classifier. The success of this decoding was used as an estimate of classification accuracy from 0 (no correct classifications) to 1 (100% correct classifications). This process was repeated 10 times and the mean ± standard deviation across the 10 folds was reported as the classification accuracy at that time. Classification accuracy of trials projected into each instantaneous subspace at 50 ms intervals was plotted as a function of trial time.”

(6) Page 9, line 268. This might be trivial, but can you speculate on why the accuracy for Instruction segments had a lower peak compared to the rest of the segments? Is it because there is less 'distinct' information embedded in neural data about the type of object manipulated until you are actually reaching toward it or holding it? The latter seems straightforward, but the former not so much.

Thank you for asking this question. We have added the following speculations (lines 592 to 604):

“Short bursts of “signal” related discharge are known to occur in a substantial fraction of PMd neurons beginning at latencies of ~60 ms following an instructional stimulus (Weinrich et al., 1984; Cisek and Kalaska, 2004). Here we found that the instantaneous subspace shifted briefly toward the subspace present at the time of instruction onset (I), similarly during execution and observation. This brief trough in principal angle (Figure 4A) and the corresponding peak in classification accuracy (Figure 7A) in part may reflect smoothing of firing rates with a 50 ms Gaussian kernel. We speculate, however, that the early rise of this peak at the time of instruction onset also reflects the anticipatory activity often seen in PMd neurons in expectation of an instruction, which may not be entirely non-specific, but rather may position the neural population to receive one of a limited set of potential instructions (Mauritz and Wise, 1986). We attribute the relatively low amplitude of peak classification accuracy for Instruction trajectory segments to the likely possibility that only the last 40 ms of our 100 ms Instruction segments captured signal related discharge.”

(7) Figure 8. Shouldn't the plots in panel A resemble those in Figure 3? Here you are projecting the hold trajectory segments into the subspace at time H, which should be the same as in Fig. 3A/B bottom right panel.

The previous Figure 8 is now Figure 8 panels A and B, and the previous Figure 3 is now Figure 5. The data used in these two figures come from two different recording sessions in two different monkeys. The current Figure 8A,B uses data from monkey F, session 2; whereas Figure 5 uses data from monkey T, session 3, which we now state in the legend to each figure, respectively. Consequently, the relative arrangement of the trajectory segments in the instantaneous subspace at time H differs. The session used in Figure 8A,B, which we now show in three dimensions, better illustrates how CCA identifies a common subspace in which execution versus observations segments show alignment (Figure 8B) that was not evident in their original subspaces (Figure 8A).

(8) Page 14, line 369. Are you computing CCA using only 2 components? I thought the subspaces were 3 dimensional. Why not align all three dimensions?

We have expanded this analysis to use all three dimensions, as illustrated in Figure 8 above.

(9) Page 14, line 407. Does this mean that instantaneous subspaces between execution and observation trials are more similar to each other during the Movement and Holding phase? Is this related to the fact that in those moments there is a smaller progressive shift of the subspaces within execution and observation trials?

Our new analyses of principal angles (see our reply to your comment 11, below) show that the progressive shifting of the instantaneous subspace continues through the movement and hold epochs. We now discuss this better alignment of the Movement and Hold trajectory segments as follows (lines 656 to 664):

“Given the complexity of condition-dependent neural trajectories across the entire time course of RGM trials (Figure 3B), rather than attempting to align entire neural trajectories, we applied canonical correlation to trajectory segments clipped for 100 ms following four well defined behavioral events: Instruction onset, Go cue, Movement onset, and the beginning of the final Hold. In all cases, alignment was poorest for Instruction segments, somewhat higher for Go segments, and strongest for Movement and Hold segments. This progressive increase in alignment likely reflects a progressive increase in the difference between average neuron firing rates for trials involving different objects (Figure 6) relative to the trial-by-trial variance in firing rate for a given object.”

(10) page 15, line 431. Typo, it should be Table 3.

We have removed Table 3 which no longer applies.

(11) A more general observation: did you try to compute another metric to assess the progressive shift of subspaces over time? I am thinking of something like computing the principal angles between consecutive subspaces. If it is true that the shifts happen over time, but it slows down during movement and hold, you should be able to conclude it from principal angles as well. Am I missing something? Is there any reason you went with classification accuracy instead of a metric like this?

Point taken. We now have calculated the principal angles as a function of time and have presented them as a new section of the Results including new Figure 4 and Figure 4 – figure supplement 3 (lines 237 to 293).

“Instantaneous subspaces shift progressively during both execution and observation

We identified an instantaneous subspace at each one millisecond time step of RGM trials. At each time step, we applied PCA to the 4 instantaneous neural states (i.e. the 4 points on the neural trajectories representing trials involving the 4 different objects each averaged across 20 trials per object, totaling 80 trials), yielding a 3-dimensional subspace at that time (see Methods). Note that because these 3-dimensional subspaces are essentially instantaneous, they capture the condition-dependent variation in neural states, but not the common, condition-independent variation. To examine the temporal progression of these instantaneous subspaces, we then calculated the principal angles between each 80-trial instantaneous subspace and the instantaneous subspaces averaged across all trials at four behavioral time points that could be readily defined across trials, sessions, and monkeys: the onset of the instruction (I), the go cue (G), the movement onset (M), and the beginning of the final hold (H). This process was repeated 10 times with replacement to assess the variability of the principal angles. The closer the principal angles are to 0°, the closer the two subspaces are to being identical; the closer to 90°, the closer the two subspaces are to being orthogonal.

Figure 4A-D illustrate the temporal progression of the first principal angle of the mirror neuron population in the three sessions (red, green, and blue) from monkey R during execution trials. As illustrated in Figure 4 – figure supplement 1 (see also the related Methods), in each session all three principal angles, each of which could range from 0° to 90°, tended to follow a similar time course. In the Results we therefore illustrate only the first (i.e. smallest) principal angle. Solid traces represent the mean across 10-fold cross validation using the 80-trial subsets of all the available trials; shading indicates ±1 standard deviation. As would be expected, the instantaneous subspace using 80 trials approaches the subspace using all trials at each of the four selected times—I, G, M, and H—indicated by the relatively narrow trough dipping toward 0°. Of greater interest are the slower changes in the first principal angle in between these four time points. Figure 4A shows that after instruction onset (I) the instantaneous subspace shifted quickly away from the subspace at time I, indicated by a rapid increase in principal angle to levels not much lower than what might be expected by chance alone (horizontal dashed line). In contrast, throughout the remainder of the instruction and delay epochs (from I to G), Figure 4B and C show that the 80-trial instantaneous subspace shifted gradually and concurrently, not sequentially, toward the all-trial subspaces that would be reached at the end of the delay period (G) and then at the onset of movement (M), indicated by the progressive decreases in principal angle. As shown by Figure 4D, shifting toward the H subspace did not begin until the movement onset (M). To summarize, these changes in principal angles indicate that after shifting briefly toward the subspace present at time the instruction appeared (I), the instantaneous subspace shifted progressively throughout the instruction and delay epochs toward the subspace that would be reached at the time of the go cue (G), then further toward that at the time of movement onset (M), and only thereafter shifted toward the instantaneous subspace that would be present at the time of the hold (H).

Figure 4E-H show the progression of the first principal angle of the mirror neuron population during observation trials. Overall, the temporal progression of the MN instantaneous subspace during observation was similar to that found during execution, particularly around times I and H. The decrease in principal angle relative to the G and M instantaneous subspaces during the delay epoch was less pronounced during observation than during execution. Nevertheless, these findings support the hypothesis that the condition-dependent subspace of PM MNs shifts progressively over the time course of RGM trials during both execution and observation, as illustrated schematically in Figure 1A.

We also examined the temporal progression of the instantaneous subspace of AE neurons. As would be expected given that AE neurons were not modulated significantly during observation trials, in the observation context AE populations had no gradual changes in principal angle (Figure 4 – figure supplement 3). During execution, however, Figure 4I-L show that the AE populations had a pattern of gradual decrease in principal angle similar to that found in the MN population (Figure 4A-D). After the instruction onset, the instantaneous subspace shifted quickly away from that present at time I and progressed gradually toward that present at times G and M, only shifting toward that present at time H after movement onset. As for the PM MN populations, the condition-dependent subspace of the PM AE populations shifted progressively over the time course of execution RGM trials.”

The related Methods are now described is subsection “Subspace Comparisons—Principal Angles”

Is there any reason you went with classification accuracy instead of a metric like this?

We now point out that (lines 295 to 297):

“The progressive changes in principal angles do not capture another important aspect of condition-dependent neural activity. The neural trajectories during trials involving different objects separated increasingly as trials progressed in time.”

And we further clarify this as follows (lines 331 to 348):

“Decodable information changes progressively during both execution and observation

As RGM trials proceeded in time, the condition-dependent neural activity of the PM MN population thus changed in two ways. First, the instantaneous condition-dependent subspace shifted, indicating that the patterns of firing-rate co-modulation among neurons representing the four different RGM movements changed progressively, both during execution and during observation. Second, as firing rates generally increased, the neural trajectories representing the four RGM movements became progressively more separated, more so during execution than during observation.

To evaluate the combined effects of these two progressive changes, we clipped 100 ms single-trial trajectory segments beginning at times I, G, M, or H, and projected these trajectory segments from individual trials into the instantaneous 3D subspaces at 50 ms time steps. At each of these time steps, we trained a separate LSTM decoder to classify individual trials according to which of the four objects was involved in that trial. We expected that the trajectory segments would be classified most accurately when projected into instantaneous subspaces near the time at which the trajectory segments were clipped. At other times we reasoned that classification accuracy would depend both on the similarity of the current instantaneous subspace to that found at the clip time as evaluated by the principal angle (Figure 4), and on the separation of the four trajectories at the clip time (Figure 5).”